# Anterior CNS expansion driven by brain transcription factors

Jesús Rodriguez Curt[1†], Behzad Yaghmaeian Salmani[1‡], Stefan Thor[1,2§]*

[1]Department of Clinical and Experimental Medicine, Linkoping University, Linkoping, Sweden; [2]School of Biomedical Sciences, University of Queensland, Saint Lucia, Australia

*For correspondence:
s.thor@uq.edu.au

Present address: [†]Department of Zoology, University of Cambridge, Cambridge, United Kingdom; [‡]Department of Cell and Molecular Biology, Karolinska Institute, Stockholm, Sweden; [§]School of Biomedical Sciences, University of Queensland, Saint Lucia, Australia

Competing interests: The authors declare that no competing interests exist.

**Abstract** During CNS development, there is prominent expansion of the anterior region, the brain. In *Drosophila*, anterior CNS expansion emerges from three rostral features: (1) increased progenitor cell generation, (2) extended progenitor cell proliferation, (3) more proliferative daughters. We find that *tailless* (mouse *Nr2E1/Tlx*), *otp/Rx/hbn* (*Otp/Arx/Rax*) and *Doc1/2/3* (*Tbx2/3/6*) are important for brain progenitor generation. These genes, and *earmuff* (*FezF1/2*), are also important for subsequent progenitor and/or daughter cell proliferation in the brain. Brain TF co-misexpression can drive brain-profile proliferation in the nerve cord, and can reprogram developing wing discs into brain neural progenitors. Brain TF expression is promoted by the PRC2 complex, acting to keep the brain free of anti-proliferative and repressive action of Hox homeotic genes. Hence, anterior expansion of the *Drosophila* CNS is mediated by brain TF driven 'super-generation' of progenitors, as well as 'hyper-proliferation' of progenitor and daughter cells, promoted by PRC2-mediated repression of Hox activity.
DOI: https://doi.org/10.7554/eLife.45274.001

## Introduction

A striking feature of the central nervous system (CNS) is the significant anterior expansion of the brain relative to the nerve cord. Anterior CNS expansion is evolutionarily conserved, becoming increasingly pronounced in vertebrates, and is particularly evident in mammals with the dramatic expansion of the telencephalon. The anterior expansion of the CNS is of importance as it likely underlies the increasingly complex behaviours governed by the CNS during evolution. However, the underlying mechanisms and genetic pathways driving anterior CNS expansion are not well understood.

*Drosophila melanogaster* (*Drosophila*) is a powerful model system for addressing this issue. The *Drosophila* larval CNS arises from ~ 1200 stem cells, neuroblasts (NBs), which delaminate from the neuroectoderm during early- to mid-embryogenesis (*Figure 1F*) (*Birkholz et al., 2013*; *Bossing et al., 1996*; *Schmid et al., 1999*; *Schmidt et al., 1997*; *Urbach et al., 2016*; *Urbach et al., 2003*; *Wheeler et al., 2009*; *Younossi-Hartenstein et al., 1996*). The CNS is segmented into 19 segments (*Birkholz et al., 2013*; *Urbach et al., 2016*; *Urbach et al., 2003*), herein referred to as brain (B1-B3), subesophageal region (S1-S3) and nerve cord (T1-T3 and A1-A10) segments (*Figure 1F*). The anterior-most brain segment, B1 (protocerebrum), displays much more extensive NB generation and contains more than twice the number of NBs found in any posterior segment (*Urbach et al., 2003*; *Younossi-Hartenstein et al., 1996*).

In the nerve cord most NBs initiate lineage progression in the Type I mode, generating daughters that divide once to generate two neurons/glia (*Doe, 2008*). Subsequently, many NBs switch to the Type 0 mode, generating directly differentiating daughters (*Baumgardt et al., 2014*; *Baumgardt et al., 2009*; *Karcavich and Doe, 2005*; *Monedero Cobeta et al., 2017*). In the brain, most NBs appear to stay in the Type I mode throughout neurogenesis and furthermore proliferate

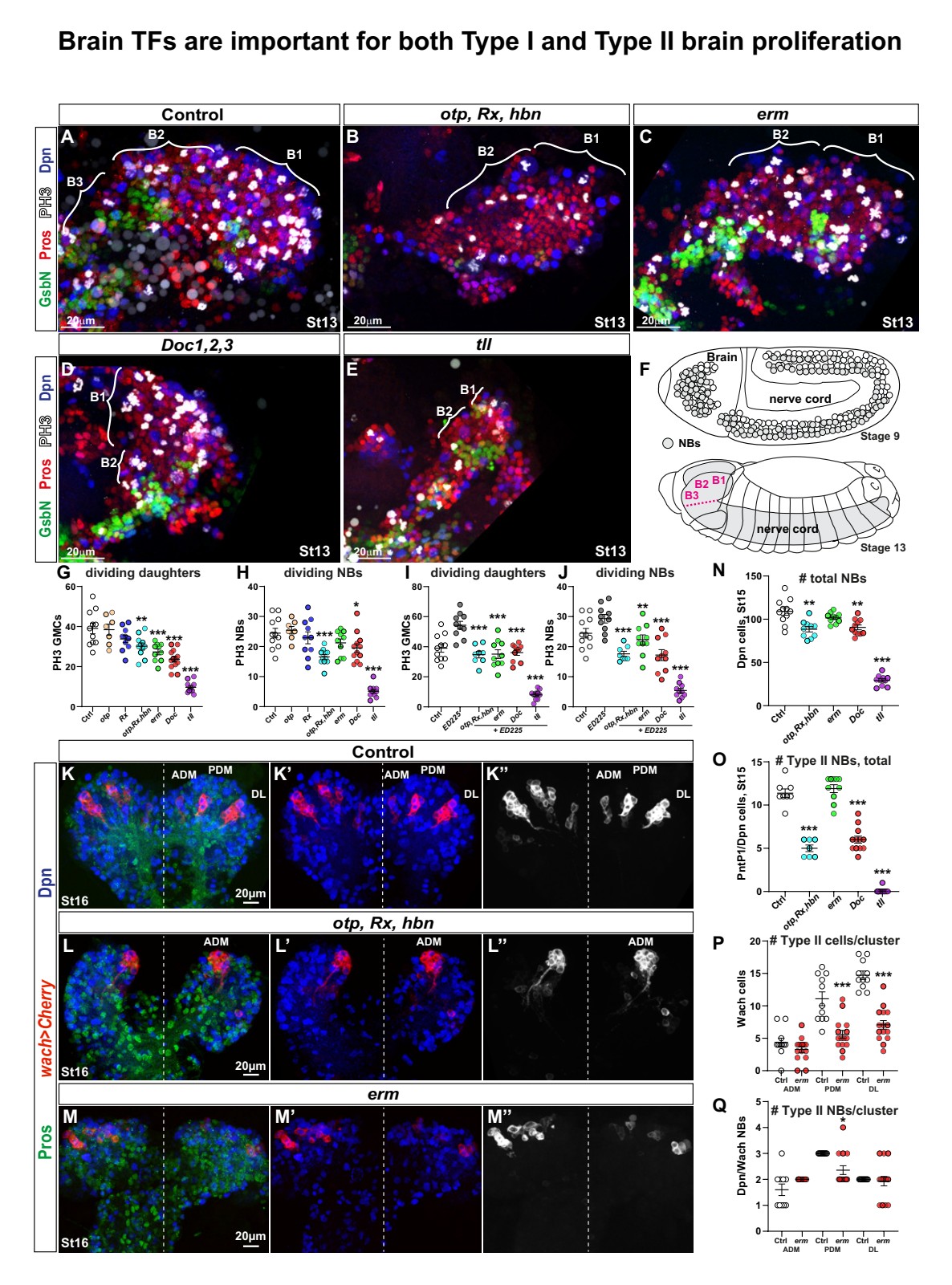

**Figure 1.** *Drosophila* brain TFs are required for proliferation and NB numbers. (A-E) Brain lobes at St13 of embryonic development, side views, anterior to the left. B1-B3 segments were delineated based on the expression of the segment-polarity marker GsbN, with a stripe of GsbN + cells marking the posterior edge of the each brain segment. PH3 labels mitotic cells. Dividing NBs are Dpn+/Pros asymmetric, while dividing daughter cells are Dpn-negative/Pros cytoplasmic. (B-E) Brain TF mutants show decreased proliferation and reduced brain size. (F) Schematic representation of the *Drosophila*

*Figure 1 continued on next page*

*Figure 1 continued*

CNS. During St8-11, the NBs are generated by delamination from the neuroectoderm, and there is a higher number of NBs in the B1 brain segment when compared to any posterior segment. By St13, NBs are undergoing lineage development, generating the brain and the nerve cord. (G-J) Quantification of dividing NBs and daughter cells in B1-B2, in control and brain gene mutants, with (G-H) or without PCD (I-J). Reduced proliferation is observed in both cases, that is when compared against wild type (G-H) or ED225 (I-J) (Student's t test; $*p \leq 0.05$, $**p \leq 0.01$, $***p \leq 0.001$; mean $\pm$ SEM; n $\geq$ 7 embryos per genotype). (K-K'') In control, *wor-Gal4 ase-Gal80, UAS-20XCherry* (*wach*) labels the three Type II NB lineage clusters: Anterior Dorso Medial (ADM), Posterior Dorso Medial (PDM) and Dorso Lateral (DL). (L-L'') In *otp,Rx,hbn* triple mutants only one Type II cluster is observed. (M-M'') In *erm* mutants all three Type II clusters are observed, but are reduced in size. (N) Quantification of total NB number in B1-B2 segments in brain gene mutants. *Otp,Rx,hbn* and *Doc1,2,3* show significant but moderate decrease while *tll* shows a dramatic reduction of NBs in B1-B2 (Student's t test; $**p \leq 0.01$, $***p \leq 0.001$; mean $\pm$ SEM; n $\geq$ 10 embryos per genotype). (O) Quantification of PntP1/Dpn positive cells in B1-B2 reveals a reduction in *Otp,Rx, hbn* and *Doc1,2,3* mutants, and a near total loss in *tll* mutants (Student's t test; $***p \leq 0.001$; mean $\pm$ SEM n $\geq$7 embryos per genotype). (P) Quantification of cell numbers in Type II (*wach*) clusters in *erm* reveals reduced lineage size for PDM and DL clusters (Student's t test; $***p \leq 0.001$; mean $\pm$ SEM; n $\geq$ 11 embryos per genotype). (Q) Quantification of NBs (Dpn+) in Type II (*wach*) clusters in control and *erm* mutants reveals a decrease in the PDM cluster (Student's t test; $*p \leq 0.05$; mean $\pm$ SEM; n $\geq$ 11 embryos per genotype). All confocal images are maximum intensity projections of multiple focal planes.
DOI: https://doi.org/10.7554/eLife.45274.002

The following source data and figure supplement are available for figure 1:

**Source data 1.** gRNA and deleted sequences for CRISPR/Cas9 deletion of *Rx,otp* and *hbn*.
DOI: https://doi.org/10.7554/eLife.45274.004
**Figure supplement 1.** Misexpression of brain TFs triggers aberrant nerve cord proliferation.
DOI: https://doi.org/10.7554/eLife.45274.003

for a longer time than NBs in the nerve cord (*Yaghmaeian Salmani et al., 2018*). Moreover, two additional and even more proliferative modes of NB behaviour exist in the B1 region: Type II NBs and mushroom body NBs (MBNB). The Type II NBs, eight in each B1 brain lobe, bud off daughter cells, denoted intermediate neural progenitor cells (INPs), which divide multiple times, budding off daughter cells that in turn divide once to generate two neurons/glia (*Álvarez and Díaz-Benjumea, 2018*; *Walsh and Doe, 2017*). The MBNBs, four in each B1 brain lobe, do not appear to bud off INPs (*Kunz et al., 2012*), and given the size of the MBNB lineages; around 30–40 cells by late embryogenesis (*Kunz et al., 2012*) it is likely that they progress in Type I mode, rather than Type 0. However, in contrast to the Type I and II NBs, and most if not all other NBs in the entire CNS, MBNBs never enter quiescence and instead continue dividing throughout embryogenesis and larvae stages. The presence of these three 'hyper-proliferative' modes of NB behaviour; the non-switching and extended proliferation Type I NBs, the Type II NBs and the never-stopping MBNBs, results in the generation of much larger average lineages in the brain at the end of embryogenesis (*Yaghmaeian Salmani et al., 2018*). When combined with the 'super-generation' of NBs in B1, this conduces to the generation of far more cells in the brain segments than in more posterior ones (*Yaghmaeian Salmani et al., 2018*).

The 'super-generation' of progenitors, specifically evident in the anterior-most brain segment (B1) is not well understood, but the head gap genes *tailless* (*tll*) and *ocelliless* (*oc*; also known as *orthodenticle* (*otd*)), are known to play a central role (*Hirth et al., 1995*; *Younossi-Hartenstein et al., 1997*). The enhanced anterior proliferation is promoted by the action of the Polycomb Repressor Complex 2 (PRC2) epigenetic complex; the key mediator of the repressive H3K27me3 mark (*Piunti and Shilatifard, 2016*; *Steffen and Ringrose, 2014*). PRC2 plays an essential role as a suppressor of anterior Hox homeotic gene expression (*Isono et al., 2005*; *Li et al., 2011*; *Struhl, 1983*; *Struhl and Akam, 1985*; *Suzuki et al., 2002*; *Wang et al., 2002*; *Yaghmaeian Salmani et al., 2018*). PRC2 thus ensures that the brain is free of Hox gene expression, thereby preventing the known anti-proliferative function of Hox genes (*Baumgardt et al., 2014*; *Karlsson et al., 2010*; *Monedero Cobeta et al., 2017*; *Prokop et al., 1998*; *Yaghmaeian Salmani et al., 2018*). However, the role of head gap genes, and other potential brain pro-proliferative genes, with regard to the brain-specific features of proliferation are not well understood. Moreover, how brain proliferation drivers intersect with the PRC2-Hox system has not been extensively addressed.

To identify genes promoting anterior CNS expansion, we applied a number of criteria to focus in on 14 transcription factors (TFs) specifically expressed in the developing embryonic brain. From these, we identified two genes and two gene families, denoted 'brain TFs' herein, which we found

to be necessary for brain NB generation and for brain-type NB and daughter cell proliferation. These include the head gap gene *tll*, as well as the *otp/Rx/hbn*, *Doc1/2/3* and *erm* genes. Brain TF co-mis-expression could drive NB and daughter cell proliferation in the nerve cord, and was sufficient to reprogram developing embryonic ectoderm and wing disc cells into brain NBs. Brain TF expression is promoted by the PRC2 complex, the main role of which is to keep the brain free of the anti-prolif-erative and repressive action of the Hox homeotic genes. Strikingly, the reduced brain proliferation observed in PRC2 mutants could be rescued by combinatorial expression of brain TFs.

These findings reveal that the super-generation of NBs in the anterior brain and the subsequent brain-specific hyper-proliferation depend upon combinatorial brain TF action. The expression of brain TFs is furthermore promoted by the PRC2 complex, which acts to keep the brain free of the anti-proliferative and repressive input of the Hox homeotic genes.

## Results

### Misexpression of brain TFs drives nerve cord proliferation

To identify transcription factors (TFs) promoting anterior CNS expansion we surveyed the literature on head gap genes and other TFs with developmental roles and/or expression in the brain (*Cohen and Jürgens, 1990*; *Hirth et al., 1995*; *Jones and McGinnis, 1993*; *Kammermeier et al., 2001*; *Kraft et al., 2016*; *Kurusu et al., 2009*; *Kurusu et al., 2000*; *Noveen et al., 2000*; *Reim et al., 2003*; *Walldorf et al., 2000*; *Weng et al., 2010*; *Younossi-Hartenstein et al., 1997*), as well as a previous head-specific transcriptome analysis (*Brody et al., 2002*) and the Berkeley *Drosophila* gene expression data base (http://insitu.fruitfly.org/cgi-bin/ex/insitu.pl). This resulted in the identification of 14 TFs expressed selectively in the brain, but broadly enough within the brain to make them candidates for substantial involvement driving anterior CNS expansion. Another limiting factor for our selection pertained to whether or not mutants and *UAS* transgenic lines were available for each gene. Of the 14 genes selected by these criteria, a subset of them were previously defined as head gap genes while others were not, and we will therefore refer to them collectively as 'brain TFs' herein. To begin addressing their role in proliferation we took a gain-of-function approach, and misexpressed the 14 brain TFs in the developing CNS, using the *elav-Gal4* driver, which expresses in NBs and daughter cells from St11-12 and onwards (*Berger et al., 2007*; *Karlsson et al., 2010*). In the control nerve cord at late stages of embryogenesis (air-filled trachea; AFT), embryonic neurogen-esis is completed and no dividing cells are observed (revealed by anti-phospho-Ser10-Histone-H3 immunostaining (PH3); *Figure 1—figure supplement 1*). In contrast, misexpression of brain TFs trig-gered aberrant mitotic cells in the stage AFT nerve cord (*Figure 1—figure supplement 1*). This anal-ysis revealed that *homeobrain*, *eyeless*, *twin of eyeless*, *brain-specific homeobox*, *empty spiracles*, *ocelliless* (also known as *orthodenticle*), *Retinal Homeobox*, and *Optix* had limited or no potency in triggering aberrant proliferation (*Figure 1—figure supplement 1*). In contrast, misexpression of four TFs triggered robust ectopic proliferation: the nuclear hormone receptor Tailless (Tll: human NR2E1/ TLX), the Prd-HD factor Orthopedia (Otp; human OTP), the T-box factor Dorsocross 2, a member of the three related and genomically adjacent Doc1/2/3 genes (human TBX2/3/6) and the Zinc-finger TF Earmuff (Erm; human FEZF1/2; *Figure 1—figure supplement 1*). While *Doc2* had strong effects, *Doc1* and *Doc3* misexpression had weaker effects (*Figure 1—figure supplement 1*). Otp is homolo-gous to Retinal homeobox (human Rax) and Homeobrain (human Arx). These three Prd-HD genes are located in the same genomic region and show genetic redundancy (see below). However, neither *UAS-Rx* nor *-hbn* transgenic lines triggered robust ectopic nerve cord proliferation (*Figure 1—figure supplement 1*). Based upon these results, we focused hereafter on the *tll* and *erm* genes, as well as the *otp/Rx/hbn* and *Doc1/2/3* gene families.

### Reduced NB and daughter cell proliferation in brain TF mutants

Next, we addressed the global proliferation effects in mutants for the brain TFs at St13. We focused on the B1-B2 region, using the B2 expression of GsbN to mark the posterior end of B2 (*Urbach, 2003*; *Yaghmaeian Salmani et al., 2018*) (*Figure 1A*). We used a combination of PH3, Dpn and Pros to distinguish mitotic NBs (Dpn+, Pros asymmetric, PH3+) from mitotic daughter cells (Dpn-negative, Pros cytoplasmic, PH3+) (*Baumgardt et al., 2014*).

For the *otp*, *Rx* and *hbn* genes, a deletion removing all three genes was available. However, individual null alleles was not available for all three genes. We therefore used CRISPR/Cas9 to generate mutants for *otp* and *Rx* (we were unsuccessful for *hbn*; see Materials and methods, and *Figure 1— source data 1*). We did not observe any global proliferation effects in *otp* or *Rx* single mutants, suggesting genetic redundancy (*Figure 1G–H*). However, using a deletion removing all three genes we observed reduced NB and daughter proliferation (*Figure 1B and G–H*). The adjacent *Doc1/2/3* genes were previously found to be redundant for amnioserosa development (*Reim et al., 2003*), and therefore we analysed a genomic deletion of all three genes. In the *Doc1/2/3* mutants, we observed global reduction of both NB and daughter cell proliferation (*Figure 1D and G–H*). *tll* also displayed significant reduction of NB and daughter cell proliferation, while *erm* only affected daughter cells (*Figure 1C, E and G–H*). To address the possibility that reduced proliferation in brain TF mutants was due to programmed cell death (PCD) we simultaneously removed PCD by using the *Df (3L)ED225* deletion, which removes *grim*, *rpr*, *skl,* and most of the upstream region of *hid*. However, in PCD/brain TF double mutants the proliferation reduction was still apparent, pointing to that the reduced NB and daughter cell proliferation was not caused by NB and daughter cell PCD (*Figure 1I–J*).

Quantification of total NB numbers revealed reduced NB numbers in all mutants but *erm* (*Figure 1N*). The loss of NBs in B1 in *tll* mutants was previously described, and is in line with its previously identified role as a head gap gene (*Cohen and Jürgens, 1991*; *Younossi-Hartenstein et al., 1997*).

The recent identification of Type II NBs in the embryonic brain and the development of markers for these NBs, that is PointedP1 and *wach-Cherry* (*wor-Gal4*, *ase-Gal80*, *UAS-Cherry*) (*Álvarez and Díaz-Benjumea, 2018*; *Neumüller et al., 2011*; *Walsh and Doe, 2017*) allowed us to selectively address development and proliferation of Type II NBs in the brain TF mutants. Using PntP1/Dpn as marker, we find a reduction of Type II NBs in *otp/Rx/hbn*, *Doc1/2/3* and *tll* mutants (*Figure 1K–L'' and O*). *otp/Rx/hbn* did however not apparently affect the ADM cluster (*Figure 1L–L'*). *erm* mutants did not display loss of type II NBs, nor an increase in Type II NBs by dedifferentiation of INPs into NBs, but rather a reduction in the number of cells generated in two of the Type II clusters; PDM and DL (*Figure 1M–M'' and O–Q*).

We conclude that the brain TFs are necessary for Type I and Type II NB generation, as well as for NB and daughter cell proliferation in the brain. Erm and Doc1/2/3 are not expressed by MBNB (see below). The role of *tll* and *Rx* in MBNB was previously addressed, showing that they are not necessary for generation of MBNB but rather their proliferation (*Kraft et al., 2016*; *Kurusu et al., 2009*).

## Expression of brain TFs in brain NBs

Next, we analysed the expression of brain TFs in the brain with respect to specific NB markers/types. In line with the reduced number of NBs generated in *tll*, *Doc1/2/3* and *otp/Rx/hbn* mutants, previous studies have revealed expression of these genes in the procephalic ectoderm (*Hartmann et al., 2001*; *Kraft et al., 2016*; *Reim et al., 2003*; *Simeone et al., 1994*; *Walldorf et al., 2000*; *Younossi-Hartenstein et al., 1997*). *erm* also shows early procephalic ectoderm expression (www.fruitfly.org), but as outlined above, this did not translate into any apparent NB generation phenotype. We used *OK107-Gal4* as marker for MBNBs (*Connolly et al., 1996*; *Kunz et al., 2012*; *Kurusu et al., 2002*), and *wach-Cherry* as marker for Type II NBs (*Álvarez and Díaz-Benjumea, 2018*). *tll* (*tll-GFP*) is expressed by all NBs in the B1 region (*Kurusu et al., 2009*; *Urbach, 2003*; *Younossi-Hartenstein et al., 1997*), and Dpn is a general NB marker.

For the *otp/Rx/hbn* gene cluster, antibodies were available to Hbn and Rx, and given the apparent genetic redundancy of these three genes (see above) we utilised these two reagents to address cluster expression. At St16, Hbn and Rx were expressed in subsets of NBs in the B1-B2 region, with Rx more broadly expressed than Hbn (*Figure 2—figure supplement 1*). Hbn and Rx expression in NBs overlapped with *tll-GFP*, and in a few cases with *erm-lacZ* (*Figure 2—figure supplement 1*). Rx was already described to be expressed in MBNBs (*Kraft et al., 2016*), and we also observed Hbn expression in MBNBs (*Figure 2—figure supplement 1*). In Type II NBs, we observed expression of both Hbn and Rx in the PDM and DL clusters, while ADM did not show expression (*Figure 2—figure supplement 2*). This is in line with the lack of apparent phenotype in the ADM cluster in *otp/Rx/hbn* triple mutants (*Figure 2L–L''*), and further indicates that *otp* is also not expressed in the ADM cluster.

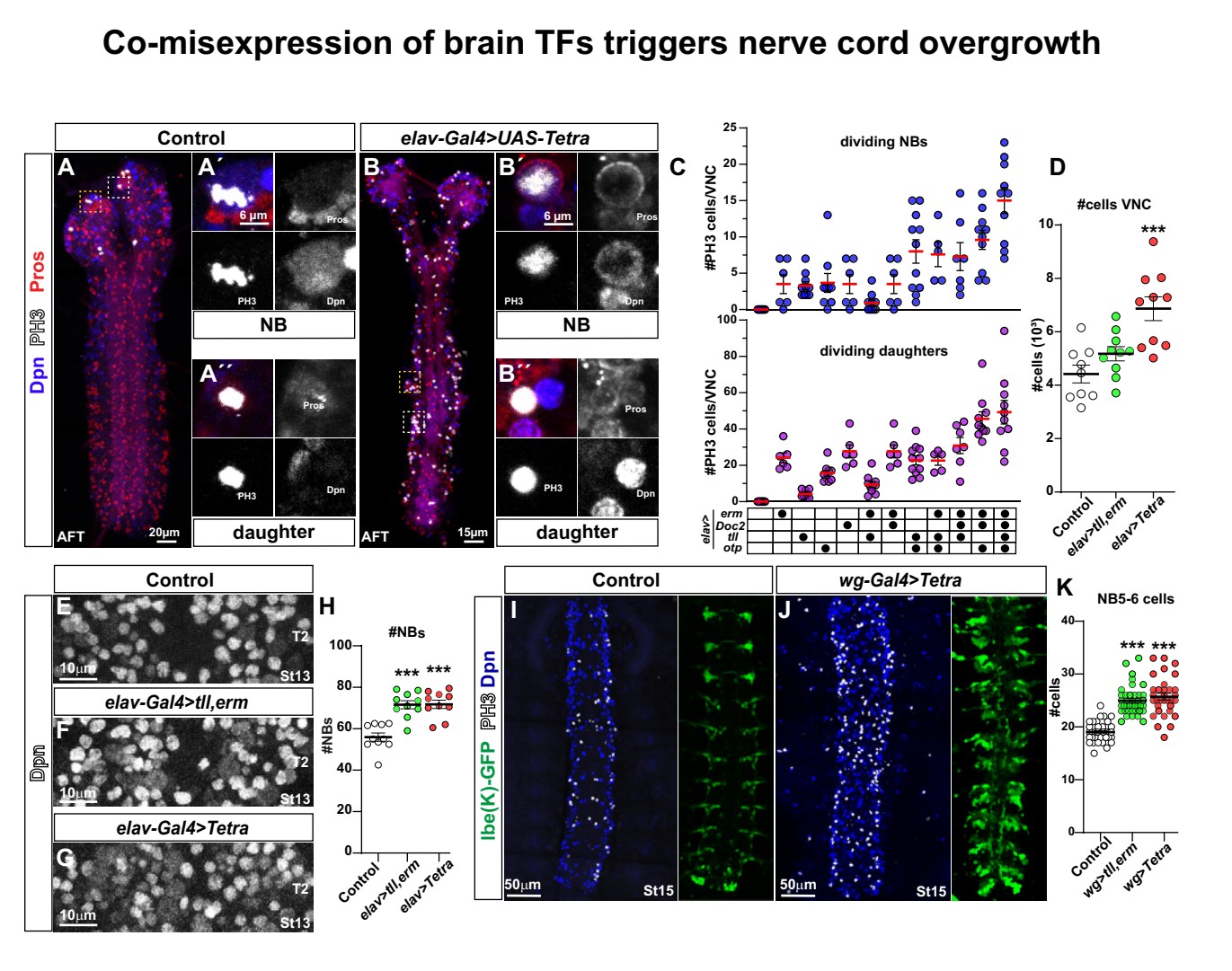

**Figure 2.** Co-misexpression of brain TFs triggers nerve cord overgrowth. (**A**) Control CNS (*elav-Gal4/+*) at AFT shows a few mitotic cells (PH3) in the brain lobes, but none in the nerve cord. (**A'**) Close-up of a mitotic NB and (**A''**) a mitotic daughter cell. (**B**) Misexpression of four brain TFs, *elav-Gal4 > UAS* Tetra (*UAS-erm, UAS-Doc2, UAS-tll, UAS-otp*), triggers aberrant proliferation in the nerve cord at AFT. (**B'**) Close-up of a mitotic NB and (**B''**) a mitotic daughter cell. (**C**) Quantitation of mitotic NBs and daughter cells per nerve cord (mean ± SEM; n ≥ 5 embryos per genotype) in control and different misexpression combinations of *UAS* transgenes. (**D**) Quantitation of total cell numbers in the nerve cord, in control (*elav-Gal4/+*), *elav > UAS tll,-erm* and *elav > UAS* Tetra (Student's t test; ***p≤0.001; mean ± SEM; n ≥ 9 embryos). (**E-G**) Dpn + cells (NBs) in control (*elav-Gal4/+*), *elav > UAS tll,-erm* and *elav > UAS* Tetra. (**H**) Quantification of NB numbers (Dpn + cells) at St13 in T2 (Student's t test; ***p≤0.001; mean ± SEM; n ≥ 10 embryos per genotype). (**I-J**) Nerve cords showing NBs (Dpn+) and the NB5-6 lineage (*lbe(K)-GFP*) at St15, in control (I; *wg-Gal4/+; lbe(K)-GFP/+*) and *wg-Gal4 > UAS* Tetra, showing an increased number of NBs (Dpn+), mitotic cells (PH3+) and an expanded NB5-6 lineage. (**K**) Quantitation of NB5-6 lineage cell numbers in the T2-T3 segments, at St15 (Student's t test; ***p≤0.001; mean ± SEM; n ≥ 10 embryos, n ≥ 32 lineages). All confocal images are maximum intensity projections of multiple focal planes. Zoomed in images are single optical sections.

DOI: https://doi.org/10.7554/eLife.45274.005

The following source data and figure supplements are available for figure 2:

**Source data 1.** Estimated average cell volume per region and genotype at AFT.
DOI: https://doi.org/10.7554/eLife.45274.010

**Figure supplement 1.** Expression of brain TFs in the developing embryo brain.
DOI: https://doi.org/10.7554/eLife.45274.006

**Figure supplement 2.** Expression of brain TFs in developing Type II NB lineages.
DOI: https://doi.org/10.7554/eLife.45274.007

**Figure supplement 3.** Brain TF co-misexpression increases EdU labelling but not cell cycle speed.
DOI: https://doi.org/10.7554/eLife.45274.008

*Figure 2 continued on next page*

*Figure 2 continued*

**Figure supplement 4.** Brain TFs control key cell cycle factor expression.

DOI: https://doi.org/10.7554/eLife.45274.009

*Doc2*, and the related and neighbouring genes *Doc1* and *Doc3*, are expressed in an overlapping manner in the embryonic procephalic region (*Reim et al., 2003*). We used a Doc2 antibody to address Doc expression. At St13, we observed Doc2 expression in a small subset of B1-B2 NBs (*Figure 2—figure supplement 1*). These NBs overlapped with *tll-GFP*, but neither with *erm-lacZ* nor Rx (*Figure 2—figure supplement 1*). We did not observe Doc2 expression in MBNBs or Type II NBs (*Figure 2—figure supplements 1* and *2*).

*erm* plays a key role in larval Type II lineages, blocking INP dedifferentiation into NBs (*Weng et al., 2010*). A genomic DNA fragment within the *erm* gene drives *Gal4* reporter expression (*GMR9D11-Gal4*) in the embryonic Type II INPs (*Walsh and Doe, 2017*), but expression of *erm* outside Type II NBs was not previously addressed. As anticipated, using *erm-lacZ* (where the same *erm GMR9D11* enhancer drives *lacZ*; *Haenfler et al., 2012*) we found that *erm* is expressed in all three Type II NB clusters; ADM, PDM and DL (*Figure 2—figure supplement 2*). Expression did not overlap with Dpn in most Type II lineages, although we did note occasional overlap (*Figure 2—figure supplement 2*). Dpn expression can persist briefly in INPs but based on their size, we distinguished Type II NBs from INPs among Dpn + cells. This indicates that, in line the larval expression, *erm* is restricted to INPs (*Figure 2—figure supplement 2*). We also observed *erm-lacZ* in a few other lineages, where expression did overlap with Dpn. We also noted expression of *tll-GFP* in all three Type II NB clusters (*Figure 2—figure supplement 2*).

Our expression analysis, together with previously published analysis, points to that most if not all Type I NBs in the B1 region express *tll* (herein; *Urbach, 2003*; *Younossi-Hartenstein et al., 1997*). In addition, many Type I NBs express Rx/Hbn. Doc2 is expressed in small subset of Type I NBs, but the global proliferation phenotype detected in *Doc1,2,3* triple mutants (*Figure 1G–H*) supports the notion of expression of Doc1 and −3 outside the Doc2 domain. All three Type II NB clusters express *tll* and *erm*, while Rx/Hbn express in PDM and DL. The MBNBs express *tll* and *Rx* (herein; *Kraft et al., 2016*; *Kurusu et al., 2009*), as well as Hbn. These expression results are in general agreement with the proliferation phenotypes.

## Combinatorial brain TF misexpression drives nerve cord proliferation

Single misexpression of the brain genes *tll*, *erm*, *Doc2* and *otp*, driven from *elav-Gal4*, triggered ectopic mitotic cells (PH3) in the nerve cord at AFT stage (*Figure 1—figure supplement 1*). To address these effects more systematically we analysed NB and daughter cell proliferation separately, and furthermore addressed the combinatorial effects of misexpression.

In control embryo nerve cords at stage AFT, we did not observe any mitotic NBs or daughter cells (*Figure 2A and C*). In contrast, single misexpression of each one of the four brain TF genes (*tll*, *erm*, *Doc2* and *otp*), driven from *elav-Gal4*, triggered aberrant mitotic NBs and daughter cells in the AFT nerve cord (*Figure 2C*). Combinatorial misexpression generally triggered stronger effects, with the co-misexpression of all four genes (*UAS-Tetra*) giving the strongest effect (*Figure 2B–C*). We focused hereafter on the *UAS-Tetra* because of its strong proliferation effect, and also on the *UAS-tll,erm* combination due to its potency in generating ectopic NBs in the wing disc (see below).

Next, we addressed the co-misexpression effects upon nerve cord size, by using 4′,6-diamidino-2-phenylindole (DAPI) nuclear staining to quantify total nuclear (cellular) volume, from which total cell number could be inferred (*Monedero Cobeta et al., 2017*; *Yaghmaeian Salmani et al., 2018*). This analysis revealed a striking increase in nerve cord volume for *UAS-Tetra*, and an upward trend for *UAS-tll,erm* (*Figure 2D*; *Figure 2—source data 1*). The increased number of cells generated as an effect of co-misexpression was also apparent at the single lineage level, evident by expanded NB5-6 lineage cell numbers (*Figure 2I–K*).

Co-misexpression of *UAS-tll,erm* or *UAS-Tetra*, driven by *elav-Gal4*, triggered more NBs in T2-T3, at St13 (*Figure 2E–H*). Because *elav-Gal4* is a late driver, activated in NBs subsequent to their delamination (*Berger et al., 2007*), the supernumerary NBs are unlikely to stem from extra NBs

generated during delamination. Rather, this effect may reflect aberrant symmetric NB divisions (see below).

Previous studies of the cell cycle length in the nerve cord revealed a cycle of ~40 min for NBs and ~100 min for daughters (*Baumgardt et al., 2014*; *Hartenstein et al., 1987*). To address the effect of brain gene misexpression of cell cycle length in the nerve cord, we pulsed St13 embryos with 5-ethynyl-2′-deoxyuridine (EdU) for 40 min, to thereby label cycling cells during S-phase (*Cappella et al., 2008*). We then stained for EdU, PH3, Pros and Dpn, allowing us to determine the S->G2- > M cell cycle speed of mitotic NBs and daughter cells (*Figure 2—figure supplement 3*). As previously observed (*Yaghmaeian Salmani et al., 2018*), we observed ratios of 10–15% of EdU/PH3 versus EdU-only labelled NBs and daughter cells in the T2 region at St13 (*Figure 2—figure supplement 3*). Co-misexpression of *UAS-tll,erm* or *UAS-Tetra* did not significantly alter this ratio (*Figure 2—figure supplement 3*). However, analysing the total number of EdU labelled NBs and daughter cells revealed significantly higher number of labelled cells in both co-misexpression cases (*Figure 2—figure supplement 3*). Hence, while *tll,erm* and *Tetra* co-misexpression cannot increase NB or daughter cell cycle speeds in the nerve cord, they do increase the number of cells in S-phase.

The strong brain TF co-misexpression effects upon proliferation in the nerve cord prompted us to analyse cell cycle factor expression. The Type I->0 daughter switch and precise NB exit both depend upon the balanced activity of four key cell cycle genes: the pro-proliferative genes *Cyclin E* (*CycE*), *E2f1*, *string* (*stg*; Cdc25) and the cell cycle inhibitor *dacapo* (*dap*; $p21^{CIP1}/p27^{KIP1}/p57^{Kip2}$) (*Baumgardt et al., 2014*; *Bivik et al., 2015*) (*Figure 2—figure supplement 4*). We analysed immunostaining intensity of these four cell cycle proteins, focusing on mitotic NBs in B1-B2 for the four brain gene mutants and in A9-A10 for the *UAS-Tetra* misexpression. Control and experimental embryos were dissected and scanned on the same slide, at St13. We found that the expression in NBs of all four cell cycle factors was affected in the mutant and co-misexpression embryos (*Figure 2—figure supplement 4*). The majority (8 out of 12) of regulatory interactions detected were logical when viewed against the mutant and misexpression phenotypes, that is cell cycle drivers such as CycE, E2f1 and Stg were often downregulated in brain TF mutants and upregulated by *UAS-Tetra* co-misexpression. Conversely, the cell cycle stopper Dap was downregulated by *UAS-Tetra* co-misexpression (*Figure 2—figure supplement 4*).

## Combinatorial brain TF misexpression reprograms the nerve cord to a brain-like CNS

To address the potency of the brain genes broadly and early in the ectoderm we turned to the *da-Gal4* driver, which expresses both maternally and ubiquitously zygotically (*Wodarz et al., 1995*). We found that co-misexpression of *UAS-tll,erm* or *UAS-Tetra* triggered extensive generation of extra NBs and daughter cells, as well as extensive proliferation, evident by Dpn, Pros and PH3 expression at St15 (*Figure 3A–C'''*). For *UAS-Tetra*, this was accompanied by an apparent thickening of the CNS (*Figure 3C'''*). To determine if the supernumerary NBs and daughter cells also underwent differentiation, we analysed expression of the Elav neuronal and Repo glia markers. This revealed striking increase in the generation of neurons for both *UAS-tll,erm* and *UAS-Tetra* (*Figure 3D–F*). In contrast, while *UAS-tll,erm* triggered ectopic glia *UAS-Tetra* displayed a striking reduction of glia (*Figure 3D–F*). The limited generation of glia in relation to neurons was reminiscent of normal brain development (*Figure 3D*), suggesting that *UAS-Tetra* may specifically trigger ectopic brain development. This notion was supported by the near complete loss of GsbN expression in *UAS-Tetra* (*Figure 3C''*), a marker normally expressed in only a few B1 NBs (*Figure 1A*) (*Urbach, 2003*; *Yaghmaeian Salmani et al., 2018*). To further test this notion, we analysed expression of the Hbn and Rx brain TFs, which revealed that while *UAS-tll,erm* showed only minor effects, *UAS-Tetra* triggered extensive ectopic Hbn and Rx expression in the nerve cord (*Figure 3G–L*).

We furthermore found that the ectopic NBs generated in the embryo, as an effect of brain TF misexpression, also expressed the NB marker Ase (*Figure 4A–C*). Intriguingly, we found that whereas control displayed very few Dpn+/Ase- expressing cells, brain TF misexpression triggered an increase in these cells (*Figure 4A–D*). The presence of Dpn+/Ase- NBs is indicative of the ectopic generation of Type II-like NBs.

Next, we addressed whether the generation of supernumerary NBs may, at least in part result from symmetric NB divisions. We turned to the more restricted *wg-Gal4* line and analysed symmetric versus asymmetric divisions in NBs and daughter cells. Interestingly, whereas control did not display

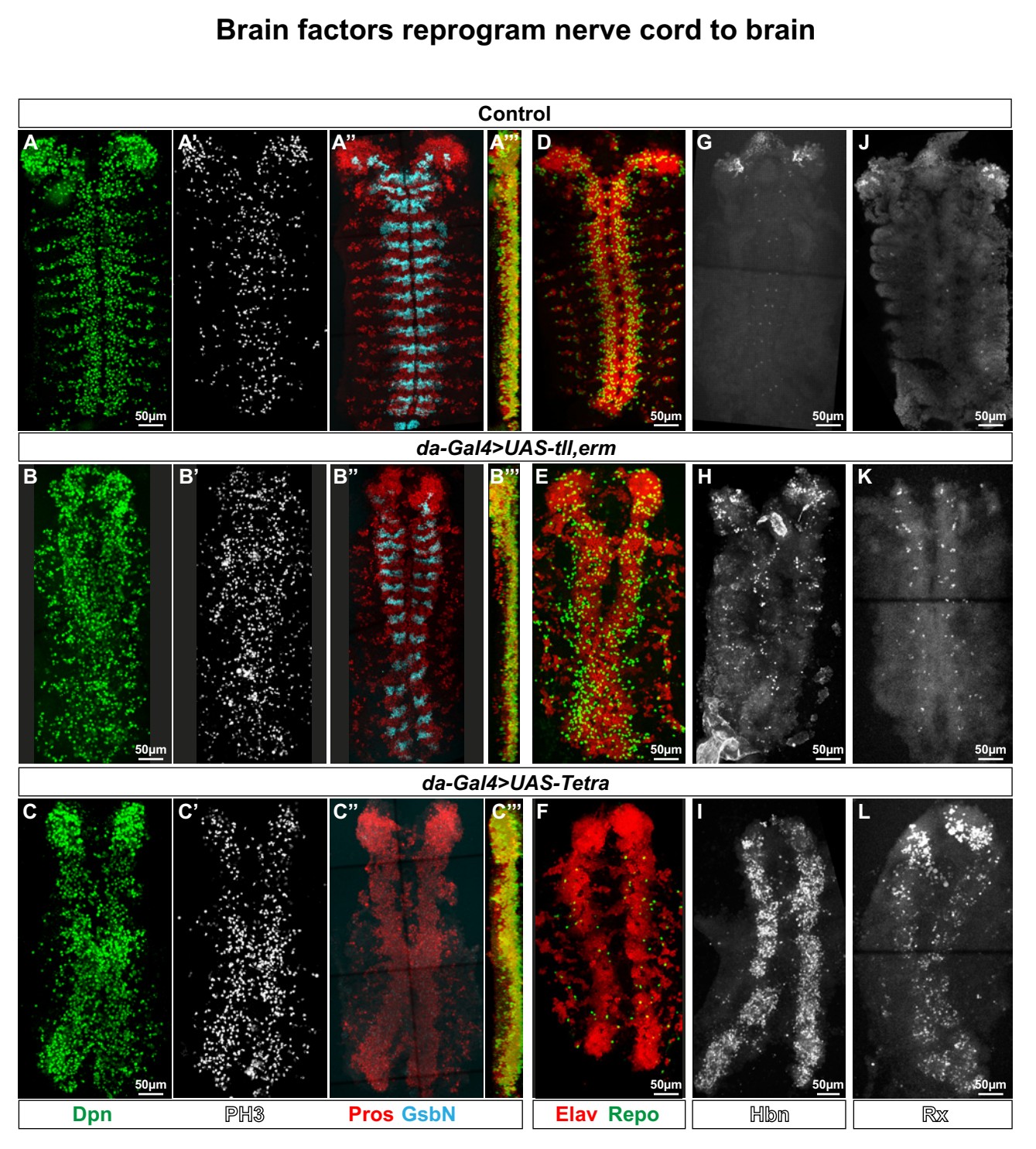

**Figure 3.** Brain TFs reprogram the nerve cord to a brain-like CNS. (A-L) Control (*da-Gal4/+*) and *da-Gal4/UAS-* co-misexpression embryo fillets, at St15, immunostained for Dpn, PH3, Pros, GsbN, Elav, Repo, Hbn or Rx. (A'''), (B''') and (C''') represent orthogonal projections of all four channels. (A-C''') *da-Gal4 > UAS tll,-erm* and *da-Gal4/UAS-Tetra* both result in generation of aberrant NBs (Dpn+), daughter cells (Pros+) and mitotic cells (PH3+). Expression of the segment polarity marker GsbN appears largely unaffected in *UAS-tll,-erm*, while *UAS-Tetra* results in widespread repression of GsbN expression in the nerve cord. (D-E) Co-misexpression of *tll,erm* triggers supernumerary neurons (Elav+) and glial cells (Repo+), while *Tetra* primarily

*Figure 3 continued on next page*

*Figure 3 continued*
generates extra neurons. (**G-L**) *da-Gal4 > UAS tll,-erm* and *da-Gal4/UAS-Tetra* both result in ectopic Hbn and Rx expression in the nerve cord, with *UAS-Tetra* showing the strongest effects. All confocal images are maximum intensity projections of multiple focal planes.
DOI: https://doi.org/10.7554/eLife.45274.011

any symmetrically dividing NBs, *wg-Gal4/UAS-tll,-erm* triggered symmetric NB divisions, with *-Tetra* also trending up (*Figure 4—figure supplement 1*). We also observed an increase in the number of symmetrically (normally) dividing daughter cells, likely as a direct effect of the increased number of daughter cell divisions (*Figure 4—figure supplement 1*).

We conclude that while *UAS-tll,erm* co-misexpression is sufficient to trigger ectopic CNS development in the embryonic ectoderm, *UAS-Tetra* co-misexpression appears to specifically trigger ectopic brain-type development. This is evident by ectopic Rx and Hbn expression, by the near complete loss of GsbN expression, by massive generation of neurons accompanied by reduction of glia below that observed in the normal VNC, and by the evidence of Type II-like NBs.

## Combinatorial brain TF misexpression reprograms wing disc to a brain-like CNS

The strong effects of brain TF misexpression in the embryonic VNC and periphery prompted us to address if they could induce NBs even in a heterologous setting. To this end, we misexpressed all 15 single, double, triple and tetra *UAS* combinations in developing wing discs; a tissue that forms as ectodermal invaginations in the embryo and grow during larval stages to form the adult wing (*Figure 5G*). We used a composite *vg-Gal4* driver that drives expression in second instar larvae wing discs and onward (*Crickmore and Mann, 2006*), and scored for ectopic NBs and daughter cells in the third instar larval discs by expression of Dpn and Pros. We observed that single misexpression of each brain TF had limited, if any, potency in generating NBs, with the only exception being *tll*, which triggered both Dpn and Pros expression (*Figure 5—figure supplement 1*). In contrast, two of the double combinations, *tll,Doc2*, and even more so, *tll,erm* triggered extensive ectopic NB and daughter cell generation (*Figure 5A–B*, *Figure 5—figure supplements 1* and *2*). Surprisingly, in the wing disc none of the triple or the Tetra combinations apparently enhanced this effect further, and rather the *Tetra* was apparently weaker in its effect than the *tll,erm* double (*Figure 5C*, *Figure 5—figure supplements 1* and *2*).

We focused our further analysis on the *tll,erm* double and the *Tetra* combinations. The reprogramming of wing disc cells into NBs, with an accompanying CNS-mode of lineage progression, was evident by the non-overlap between Dpn and Pros expressing cells (NBs and daughter cells) (*Figure 5A–C*). To further confirm that these ectopic progenitors were NBs, rather than sensory organ progenitors (SOPs), we used the temporal NB factor Cas (*Baumgardt et al., 2014*; *Baumgardt et al., 2009*; *Isshiki et al., 2001*; *Kambadur et al., 1998*) and the CNS-specific axon marker BP102 (*Elkins et al., 1990*), neither of which are expressed by wing disc SOPs (*Figure 5—figure supplement 3*) (*Bahrampour et al., 2017*). While *UAS-tll,erm* triggered ectopic expression of both Cas and BP102, *UAS-Tetra* triggered activation only of Cas, and to a lesser extent (*Figure 5H*, *Figure 5—figure supplement 3*). We also observed that *UAS-tll,erm* triggered ectopic expression of the asymmetric factors Mira and Insc in NBs (Dpn) (*Figure 5H*, *Figure 5—figure supplement 3*), as well as non-overlapping expression of Elav (neurons) and Repo (glia) (*Figure 5H*, *Figure 5—figure supplement 4*). Again, the *UAS-Tetra* had weaker effects, and only triggered Mira and Elav, but not Insc and Repo (*Figure 5H*, *Figure 5—figure supplements 3* and *4*).

Orthogonal optical sections furthermore revealed that disc cells that were reprogrammed into NBs delaminated from the epithelial plane, akin to NB delamination observed during normal embryonic neurogenesis (*Figure 5—figure supplement 5*).

Similar to the results in the embryo we found that the ectopic NBs generated in the wing discs also expressed the NB marker Ase (*Figure 5—figure supplement 5*). Intriguingly, Ase expressing cells only partly overlapped with Dpn (*Figure 5—figure supplement 4*). Thus, similar to the embryonic misexpression results, the presence of Dpn+/Ase-negative NBs is indicative of the ectopic generation of Type II-like NBs.

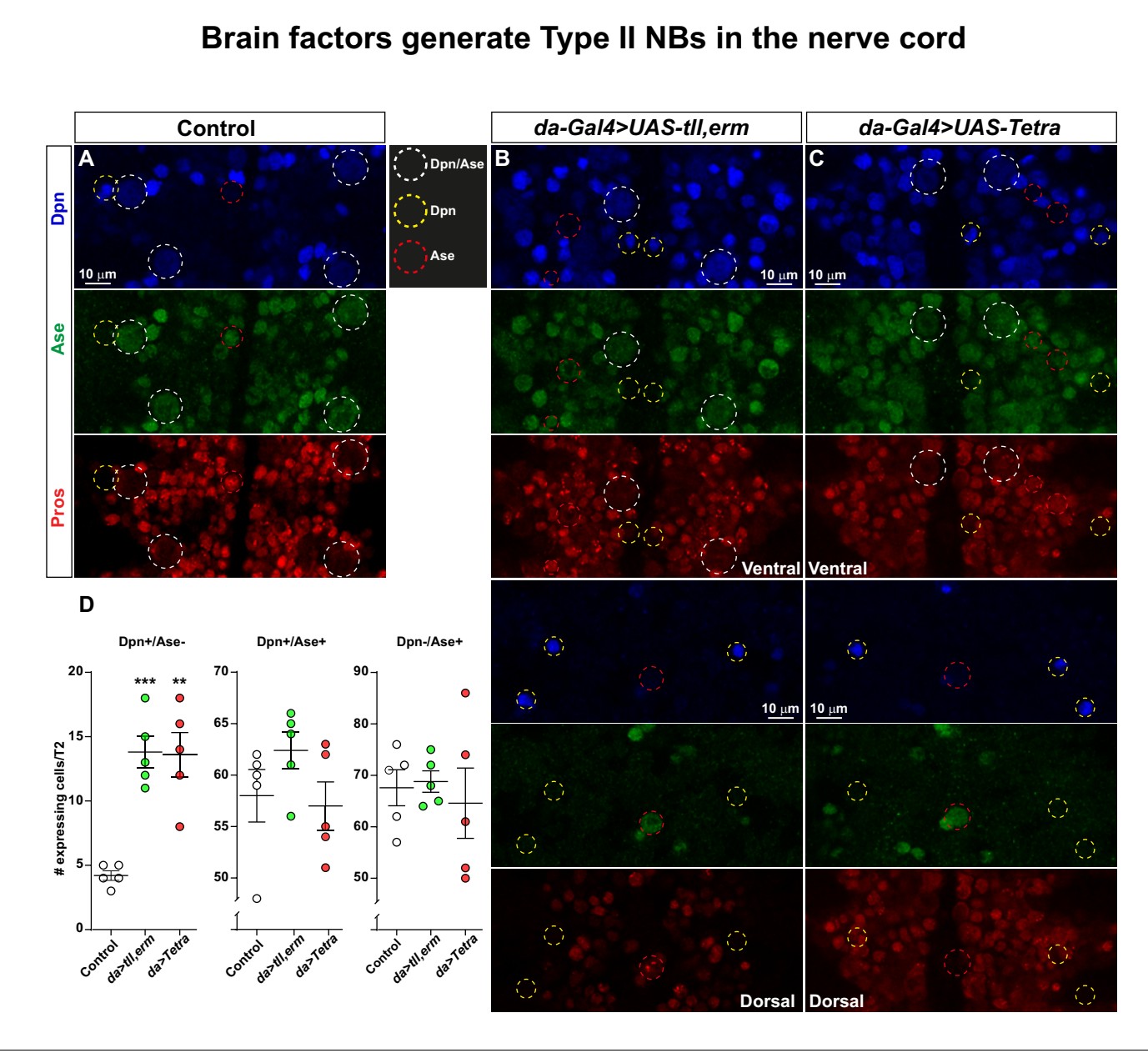

**Figure 4.** Brain TFs trigger generation of Type II-like NBs in the nerve cord. (**A-C**) Control (*da-Gal4/+*) and *da-Gal4/UAS-* co-misexpression embryo fillets, at St13, stained for Dpn, Ase and Pros. (**D**) Quantification of Dpn-only, Dpn/Ase and Ase-only expressing cells in control and misexpression, revealing significant increase in Dpn-only NBs in the co-misexpression embryos (Student's t test; **p≤0.01, ***p≤0.001; mean ± SEM; n ≥ 5 embryos). Confocal images are maximum intensity projections of multiple focal planes. Zoomed in images are single optical sections.
DOI: https://doi.org/10.7554/eLife.45274.012

The following figure supplement is available for figure 4:

**Figure supplement 1.** Brain TFs trigger symmetric NB divisions.
DOI: https://doi.org/10.7554/eLife.45274.013

To address if co-misexpression could trigger brain-type NB generation, we again turned to the Rx and Hbn brain TFs, and indeed observed ectopic expression of these markers, albeit to a lesser extent than Dpn (*Figure 5D–F*).

These results demonstrate that co-misexpression of the brain TFs can robustly trigger ectopic generation of NBs in the developing wing discs, which by several criteria mimic embryonic NBs and

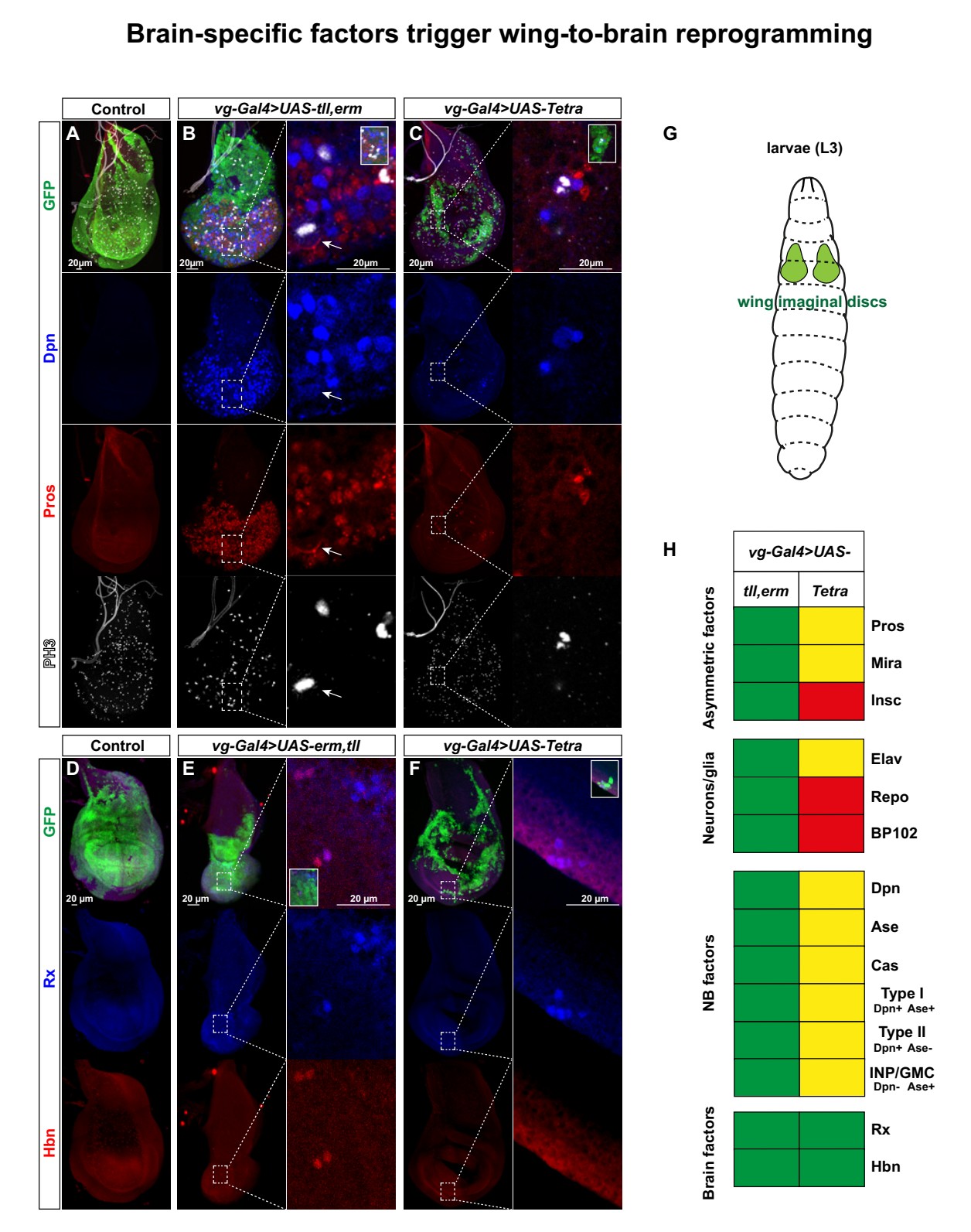

**Figure 5.** Brain TFs trigger wing-to-brain reprogramming. (**A**) Control (*vg-Gal4, UAS-GFP/+*) L3 larval imaginal wing discs do not display NBs, evident by lack of Dpn and Pros expressing cells. (**B-C**) *vg-Gal4, UAS-GFP > UAS tll, erm* or *UAS-Tetra* triggers ectopic NBs and daughter cell specification (arrow in B point to an asymmetrically dividing NB). (**D**) Control wing discs do not display expression of the brain markers Rx and Hbn. (**E-F**) *vg-Gal4, UAS-GFP > UAS tll,-erm* or *UAS-Tetra* triggers ectopic expression of Rx and Hbn. (**G**) Cartoon showing the developing wing imaginal discs in the L3

*Figure 5 continued on next page*

*Figure 5 continued*
larvae. (H) Summary of the co-misexpression effects, based upon this figure, as well as figures S7-S10. In contrast to the embryonic misexpression effects (*Figures 2* and *3*), in the wing imaginal disc *tll,erm* shows a more complete NB reprogramming effect that *Tetra* (green = strong effect; yellow = intermediate effect; red = no effect; Pros is expressed in both *UAS* combinations, but only asymmetric in *tll,erm*). Confocal images of entire wing discs are maximum intensity projections of multiple focal planes. Zoomed in images are single optical sections.
DOI: https://doi.org/10.7554/eLife.45274.014
The following figure supplements are available for figure 5:

**Figure supplement 1.** Brain TFs trigger wing disc neural reprogramming.
DOI: https://doi.org/10.7554/eLife.45274.015
**Figure supplement 2.** Brain TFs trigger wing disc neural reprogramming.
DOI: https://doi.org/10.7554/eLife.45274.016
**Figure supplement 3.** Brain TFs trigger wing disc neural reprogramming.
DOI: https://doi.org/10.7554/eLife.45274.017
**Figure supplement 4.** Brain TFs trigger wing disc neural reprogramming.
DOI: https://doi.org/10.7554/eLife.45274.018
**Figure supplement 5.** Brain TFs trigger NB delamination in wing discs.
DOI: https://doi.org/10.7554/eLife.45274.019

their ensuing lineages. The ectopic expression of the brain TFs Rx and Hbn further indicate the generation of brain-type NBs. Moreover, the presence of Dpn+/Ase-negative NBs indicates the generation of Type II-like NBs. Surprisingly, in contrast to the embryo effects, in the wing discs *UAS-tll,erm* was more efficient in all aspects of reprogramming than *UAS-Tetra* (*Figure 5H*).

## Brain TFs repress Hox homeotic expression and vice versa

Previous studies have revealed that the Hox homeotic genes act in an anti-proliferative manner in the nerve cord (*Prokop et al., 1998*), and promote the Type I->0 daughter cell switch and NB cell cycle exit (*Baumgardt et al., 2014*; *Karlsson et al., 2010*; *Monedero Cobeta et al., 2017*). Therefore, we sought to determine if the nerve cord-to-brain-like CNS reprogramming, driven by brain TF co-misexpression, was accompanied by reduction of Hox gene expression. To this end, we focused on the three Hox genes of the *Bithorax Complex* (*BX-C*); *Ultrabithorax* (*Ubx*), *abdominal-A* (*abd-A*) and *Abdominal-B* (*Abd-B),* acting in abdominal segments (A1-A10) (*Hirth et al., 1998*; *Monedero Cobeta et al., 2017*; *Prokop et al., 1998*). Indeed, we found that *UAS-tll,erm* and *UAS-Tetra* repressed Ubx and Abd-B expression, while only *UAS-tll,erm* repressed Abd-A, in the highest-expressing abdominal segment for each Hox factor (*Monedero Cobeta et al., 2017*) (*Figure 6A–J*).

Conversely, we analysed brain TFs expression in triple Hox (*BX-C*) co-misexpressing embryos, and observed down-regulation of Bsh, Hbn and Rx in the B1-B2 NBs (*Figure 6—figure supplement 1*). Previous studies found that *BX-C* co-misexpression caused repression of Tll expression (*Yaghmaeian Salmani et al., 2018*). Hence, brain TFs and BX-C Hox genes can repress each other when misexpressed in each other's expression domain.

## PRC2 ensures brain TFs expression

Mutation of the key PRC2 component *extra sex combs* (*esc*; mammalian EED) results in a complete loss of the repressive histone mark H3K27me3 in the embryo (*Yaghmaeian Salmani et al., 2018*). *esc* mutants display anterior expansion of embryonic Hox gene expression (*Struhl, 1983*; *Struhl and Akam, 1985*), including ectopic expression of the *BX-C* Hox genes into the B1 segment (*Yaghmaeian Salmani et al., 2018*). This triggers reduced brain NB and daughter cell proliferation, and conversion of the average brain lineage size to that normally observed in the thoracic ones (*Yaghmaeian Salmani et al., 2018*). Brain TF expression (Tll and Doc2) was also down-regulated in NBs of *esc* mutants (*Yaghmaeian Salmani et al., 2018*). We analysed the expression of Rx in the *esc* maternal and zygotic mutants, and observed reduction also of Rx expression in NBs (*Figure 6O*).

The anterior expansion of Hox gene expression into the brain in *esc*, the reduction of Tll, Doc2 and Rx expression, and the repressive interplay between brain genes and Hox genes prompted us to attempt to rescue *esc* with *UAS-tll,erm* and *UAS-Tetra*. As previously described (*Yaghmaeian Salmani et al., 2018*), in *esc* maternal/zygotic mutants, at St13, the brain displays reduced proliferation of both NBs and daughter cells (*Figure 6K–L and P*). Strikingly, misexpression

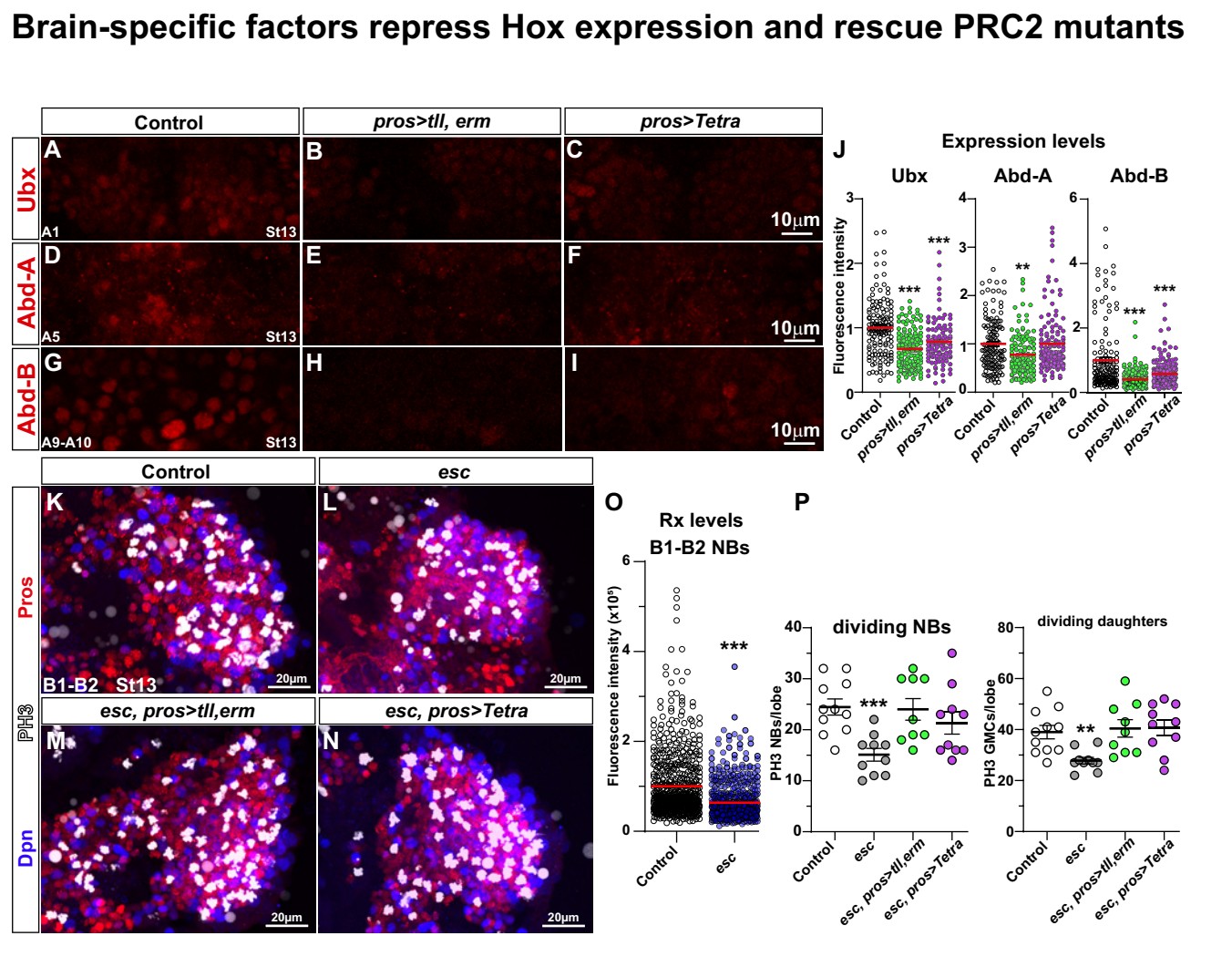

**Figure 6.** Brain TFs repress Hox expression and rescue PRC2 mutants. (A-I) Control (*da-Gal4/+*) and *da-Gal4/UAS-* co-misexpression embryo nerve cords, at St13, stained for Ubx, Abd-A or Abd-B, in the highest-expressing abdominal segment for each Hox factor (A1 for Ubx; A5 for Abd-A; A8-A10 for Abd-B) (*Monedero Cobeta et al., 2017*). (J) Quantification of Ubx, Abd-A and Abd-B expression levels in NBs in the same respective segments (Student's t test; **p≤0.01, ***p≤0.001; mean ± SEM; n ≥ 3 embryos, n ≥ 108 NBs). (K-N) Control (*OrR*), *esc* maternal/zygotic mutants, and *esc* maternal/zygotic mutants co-expressing *tll,erm* (*pros > tll, erm*) or Tetra (*pros > Tetra*), stained for Dpn, Pros and PH3. In *esc* maternal/zygotic mutants, proliferation is reduced. This phenotype can be rescued by either *UAS-tll,-erm* or *UAS-Tetra* expression. (O) Quantification of Rx expression in B1-B2 NBs, in control and *esc* mutants, reveal significant reduction of Rx in *esc* mutants (Mann-Whitney U-test; ***p≤0.001; mean ± SD; n ≥ 6 brain lobes, n ≥ 620 NBs per genotype). (P) Quantitation of NB and daughter cell proliferation (Student's t test; **p≤0.01, ***p≤0.001; mean ± SD; n ≥ 9 embryos per genotype; note that control data in P are reproduced for reference from *Figure 1G–H*). (A-I) Single optical sections. (K-N) Maximum intensity projections of multiple focal planes.

DOI: https://doi.org/10.7554/eLife.45274.020

The following figure supplement is available for figure 6:

**Figure supplement 1.** Hox genes repress brain TFs.

DOI: https://doi.org/10.7554/eLife.45274.021

of *tll,erm* or *Tetra*, driven by *pros-Gal4*, could fully rescue the reduced proliferation observed in *esc* mutants (*Figure 6M–N and P*). Hence, the proliferative effects observed in maternal/zygotic *esc* mutants, which displays complete loss of the repressive H3K27me3 histone modification and anterior expression of Hox genes, could be fully rescued by transgenic expression of brain TFs.

## Discussion

### Brain TFs promote anterior CNS expansion by driving super-generation of NBs

Detailed analysis of *Drosophila* CNS development has revealed that there is 'super-generation' of NBs in the B1 segment;~160 NBs in B1 compared to 28–70 NBs/segment for each of the 18 posterior segments (B2-A10) (*Álvarez and Díaz-Benjumea, 2018*; *Birkholz et al., 2013*; *Bossing et al., 1996*; *Schmid et al., 1999*; *Schmidt et al., 1997*; *Urbach et al., 2016*; *Urbach et al., 2003*; *Walsh and Doe, 2017*; *Wheeler et al., 2009*; *Younossi-Hartenstein et al., 1996*). In the ventral neurogenic regions (generating the nerve cord) a single NB delaminates from each proneural cluster (*Campos-Ortega, 1993*). In contrast, the NB super-generation in B1 stems, at least in part, from group delamination of NBs (*Schmidt-Ott and Martin Technau, 1994*; *Technau and Campos-Ortega, 1985*; *Urbach et al., 2003*). The specification of NB cell fate depends upon low, or no, Notch activity (*Campos-Ortega, 1995*; *Skeath, 1999*; *Skeath and Thor, 2003*). In line with this notion, evidence points to reduced Notch signalling in the procephalic neuroectoderm (*Stuttem and Campos-Ortega, 1991*; *Urbach et al., 2003*).

Head gap genes, such as *tll*, were previously shown to be important for B1 NB generation (*Younossi-Hartenstein et al., 1997*), and in line with this we observed strikingly reduced NB generation in *tll*. Does *tll* intersect with Notch signalling? *tll* mutants show loss of expression of the proneural gene *l'sc* (*Younossi-Hartenstein et al., 1997*), which is negatively regulated by Notch (*Skeath and Carroll, 1992*). Recent studies furthermore reveal an intimate interplay between *tll* and Notch signalling in the developing *Drosophila* embryonic optic placodes (*Mishra et al., 2018*). In addition, the *C. elegans tll* orthologue *nhr-67* regulates both *lin-12* (Notch) and *lag-2* (Delta) during uterus development (*Verghese et al., 2011*). Strikingly, in the mouse brain, the *tll* orthologue *Nr2E1* (aka *Tlx*) was recently shown to negatively regulate the canonical Notch target gene *Hes1* (*Luque-Molina et al., 2019*). Against this backdrop, it is tempting to speculate that the group NB delamination normally observed in the procephalic region results, at least in part from *tll* repression of the Notch pathway. Indeed, *tll* was the only one of the four TFs that could act alone to trigger ectopic NBs in the wing disc.

Other previously identified head gap genes are *oc* (also known as *orthodenticle: otd*), *buttonhead* (*btd*) and *ems* (*Cohen and Jürgens, 1991*; *Reichert and Simeone, 1999*; *Thor, 1995*). However, we did not observe that misexpression of *oc* or *ems* from *elav-Gal4* efficiently drove ectopic proliferation in the nerve cord. Moreover, *oc* acts both in B1-B2, *ems* in B2-B3 (*Hirth et al., 1995*), being repressed from B1 by *tll* (*Hartmann et al., 2001*), and *btd* acts in B2-B3 (*Cohen and Jürgens, 1990*; *Younossi-Hartenstein et al., 1997*). Because B2 and B3 segments do not display super-generation of NBs (*Urbach et al., 2003*) these findings point to *tll* as the key head gap gene driving the super-generation of NBs specifically observed in the B1 segment.

We furthermore observed reduced NB generation in the triple *otp/Rx/hbn* and *Doc1/2/3* mutants. This would tentatively place them in the category of head gap genes, at least as far as being important for NB generation. However, their effects on NB generation is weaker than that observed in *tll* mutants. In addition, *otp/Rx/hbn* and *Doc1/2/3* show genetic redundancy [herein; (*Reim et al., 2003*)]. The combination of genetic redundancy and their weaker effects on NB generation, likely explain why they were not previously categorised as head gap genes.

The connection between the brain TFs studied herein and NB super-generation is not only evident from the mutant phenotypes, but also from their potent gain-of-function effects. Strikingly, we find that brain TF co-misexpression was sufficient to generate ectopic NBs in the embryonic ectoderm and developing wing discs. A number of markers indicate that these ectopic NBs undergo normal CNS NB lineage progression, generating neurons and glia. Moreover, the ectopic expression of the brain-specific factors Rx and Hbn, the apparently higher neuron/glia ratio, the reduced GsbN expression, the generation of Dpn+/Ase- NBs (Type II-like) in both the embryonic ectoderm and wing discs, in combination suggest that brain TF co-misexpression specifically triggered reprogramming towards a B1 brain-like phenotype.

One surprising finding pertains to the clear difference between the potency of the *tll,erm* double and the *Tetra* in the embryonic ectoderm versus the wing disc, with the double being more potent in the wing disc and the *Tetra* more potent in the embryo. Indeed, in the wing disc the strong effect of *tll,erm* is suppressed by the addition of any combination of *otp* and *Doc2*. There is no obvious

explanation for the different responsiveness to brain TF misexpression in the two tissues, but it may reflect the fact the embryonic neuroectoderm is already primed for the generation of NBs.

Another surprising finding pertains to the role of *erm* in embryonic versus larva Type II NBs. Previous studies of *erm* function in the larvae found that *erm* mutants displayed more Type II NBs. Larval MARCM clone induction and marker analysis demonstrate that this is due to de-differentiation of INPs back to type II NBs, rather than excess generation of Type II NBs in the embryo (*Weng et al., 2010*). We did not find extra Type II or Type I NBs in *erm* mutants but rather reduced number of cells generated in the embryonic Type II lineages, showing that *erm* is important for lineage progression. Hence, the role of *erm* appears to be different in the embryonic versus larval Type II lineages.

## Brain TFs promote anterior CNS expansion by driving NB and daughter cell proliferation

In addition to the NB super-generation in B1, recent studies reveal that three different lineage topology mechanisms underlie the hyper-proliferation of the brain. First, the majority of NBs (136 out 160 NB) display a protracted phase of NB proliferation, and do not show evidence of switching from

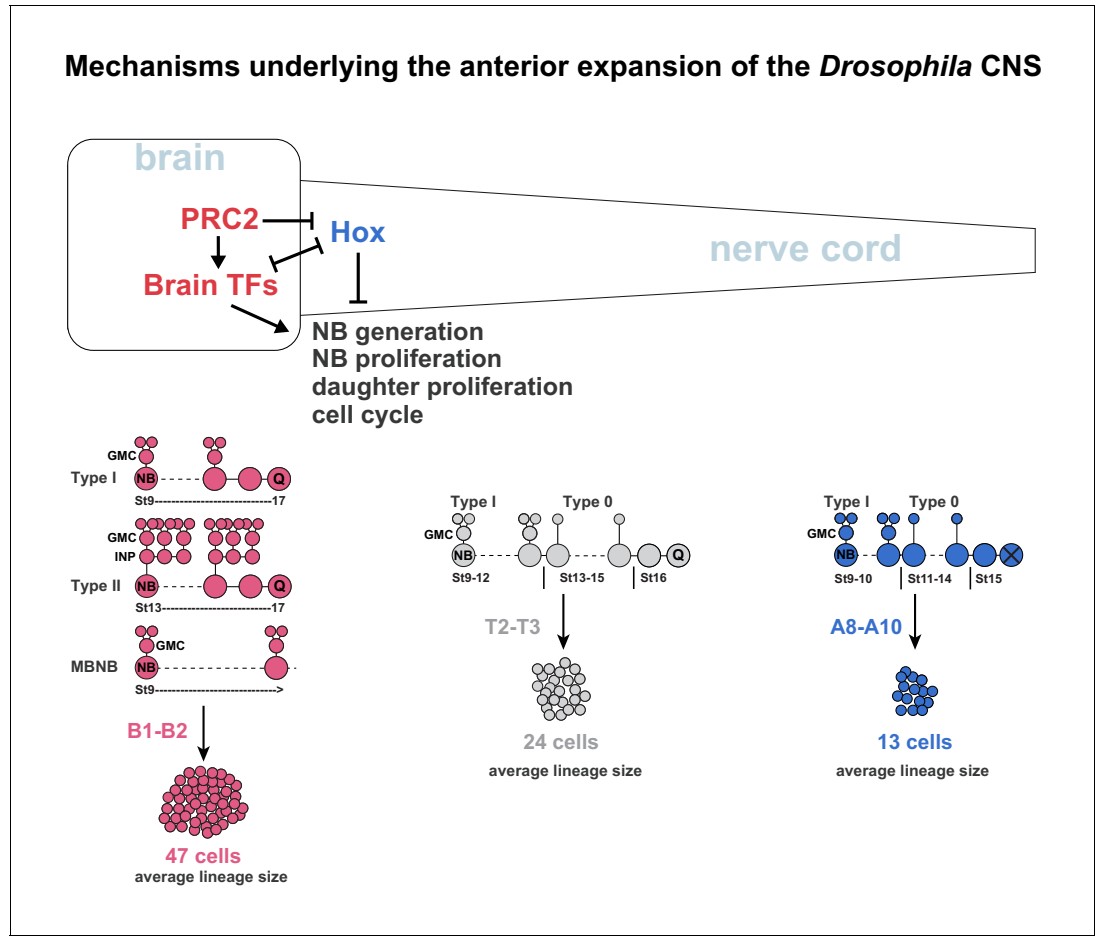

**Figure 7.** Mechanisms underlying the anterior expansion of the *Drosophila* CNS. Brain TFs act in two manners to promote anterior CNS expansion. First, brain TFs drive the super-generation of NBs observed in the B1 brain segment. Second, brain TFs promote the three different hyper-proliferative lineage progression modes observed in the brain; Type I, Type II and MBNB, which manifest as an extended phase of NB proliferation and more elaborate daughter cell proliferation. In the nerve cord, Hox genes act to limit NB and daughter cell proliferation, resulting in a switch from Type I to Type 0 proliferation, and an earlier NB cell cycle exit. The distinct lineage profiles evident in the brain versus nerve cord result in dramatically different average lineages sizes along the A-P axis. The combination of NB super-generation and the hyper-proliferative lineage modes results in vastly more cells generated in the brain B1 segments than in the nerve cord segments (*Monedero Cobeta et al., 2017*; *Yaghmaeian Salmani et al., 2018*). PRC2 acts to prevent Hox gene expression in the brain, thereby promoting brain TF expression and thereby anterior CNS expansion.
DOI: https://doi.org/10.7554/eLife.45274.022

Type I to Type 0 daughter proliferation (*Yaghmaeian Salmani et al., 2018*) (*Figure 7*). Second, the eight MBNBs, which appear to divide in the Type I mode and never enter quiescence, also generate large lineages (*Kunz et al., 2012*) (*Figure 7*). Third, the 16 Type II NBs progress by budding off INP daughter cells, which divide multiple times to generate daughter cells that in turn divide once, hence resulting in lineage expansion (*Álvarez and Díaz-Benjumea, 2018*; *Walsh and Doe, 2017*) (*Figure 7*). In contrast, in the nerve cord many NBs switch from Type I to Type 0 (*Baumgardt et al., 2014*; *Baumgardt et al., 2009*; *Karcavich and Doe, 2005*; *Monedero Cobeta et al., 2017*), and all halt neurogenesis by mid-embryogenesis (*Baumgardt et al., 2014*; *Prokop et al., 1998*) (*Figure 7*). The Hox anti-proliferation gradient further results in a gradient of the Type I->0 switch and NB exit along the nerve cord (*Monedero Cobeta et al., 2017*). The combined effects of these alternate lineage topology behaviours translate into striking differences in the average lineage size in the brain when compared to the nerve cord (*Yaghmaeian Salmani et al., 2018*) (*Figure 7*). Moreover, the three different modes of more extensive NB and daughter cell proliferation combine with the super-generation of NBs in B1 to generate many more cells in the B1 brain segment, when compared to all posterior segments (*Figure 7*) (*Monedero Cobeta et al., 2017*; *Yaghmaeian Salmani et al., 2018*).

The brain TFs studied herein are expressed in several or all (Tll) of the three brain NB types, and are important for both NB and daughter cell proliferation [herein; (*Kraft et al., 2016*; *Kurusu et al., 2009*). In line with this, brain TF ectopic expression, with the late neural driver *elav-Gal4*, drives aberrant nerve cord proliferation and blocks both the Type I->0 daughter cell proliferation switch and NB cell cycle exit. This results in the generation of supernumerary cells, evident both by the expansion of specific lineages and an increase in overall nerve cord cell numbers. We find that both the double and Tetra misexpression can trigger the ectopic generation of what appears to be a mix of Type I and Type II-like NBs. The mix of these two NB types may reflect that the misexpression scenario does not accurately and reproducibly recreate the temporal order of the brain TFs, with for example *tll* expressed prior to *erm* in the wild type.

The ectopic appearance of symmetrically dividing NBs in the brain TF co-misexpression nerve cords is more difficult to explain. However, since there normally are divisions of cells in the neuroectodermal layer prior to NB delamination, and given the early expression of the brain TFs (prior to NB delamination), it is tempting to speculate that brain TF co-misexpression to some extent can trigger an early neuroectodermal cell fate.

We recently found that NB and daughter proliferation is also promoted by a set of early TFs expressed by most, if not all NBs (*Bahrampour et al., 2017*). Strikingly, these TFs are expressed at higher levels in the brain, due to the lack of Hox expression therein, thereby contributing to the extended NB proliferation and more proliferative daughter cells observed in the brain (*Bahrampour et al., 2019*). It will be interesting to address the possible regulatory interplay between these broadly expressed early NB factors and the brain TFs described herein.

## PRC2 promotes the separation of brain TF and Hox gene expression into brain and nerve cord, respectively

Gene expression studies have revealed the mutually exclusive territory of brain TF and Hox gene expression in the *Drosophila* CNS (*Reichert and Simeone, 1999*; *Sprecher and Reichert, 2003*; *Thor, 1995*). In line with this notion, we found that co-misexpression of brain TFs in the nerve cord repressed expression of the posterior Hox genes of the *BX-C*, and conversely that *BX-C* co-misexpression repressed several brain TFs; Bsh, Rx, Hbn, Tll and Doc2 [herein; (*Yaghmaeian Salmani et al., 2018*).

A key 'gate-keeper' of the brain versus nerve cord territories appears to be the PRC2 epigenetic complex. Removing PRC2 function results in complete loss of the H3K27me3 repressive epigenetic mark and anterior expansion of the expression of all Hox genes (*Isono et al., 2005*; *Li et al., 2011*; *Struhl, 1983*; *Struhl and Akam, 1985*; *Suzuki et al., 2002*; *Wang et al., 2002*; *Yaghmaeian Salmani et al., 2018*). This furthermore results in repression of brain TF expression, that is Tll and Doc2 (*Yaghmaeian Salmani et al., 2018*), as well as Rx (herein). Surprisingly, in spite of the many roles that PRC2 may play, we found that transgenic brain TF co-expression could rescue the PRC2 mutant proliferation defects. Given the repressive action of BX-C Hox genes on brain TFs, this suggests that the principle role of PRC2 during early CNS development, at least regarding proliferation, is to ensure that Hox genes are prevented from being expressed in the brain, thus ensuring brain TF expression. Indeed, we recently demonstrated that the reduced brain proliferation

observed in *esc* mutants could also be fully rescued by the simultaneous removal of the posterior-most and most anti-proliferative Hox gene, *Abd-B* (*Bahrampour et al., 2019*).

## Brain versus nerve cord; evolutionarily conserved distinct modes of neurogenesis and genetic control?

In mammals, the precise number of neural progenitors present at different axial levels during embryonic development has not yet been mapped. However, the wider expanse of the anterior embryonic neuroectoderm (*Shimamura et al., 1995*) would suggest the generation of more progenitors anteriorly. There is also an extended phase of neurogenesis in the forebrain, when compared to the spinal cord (*Caviness et al., 1995*; *Huang et al., 2013*; *Kicheva et al., 2014*; *Yaghmaeian Salmani et al., 2018*). Dividing daughter cells (most often referred to as basal progenitors; bP) have been identified along the entire A-P axis of the mouse CNS (*Haubensak et al., 2004*; *Smart, 1972a*; *Smart, 1972b*; *Smart, 1973*; *Smart, 1976*; *Tarabykin et al., 2001*; *Wang et al., 2011*). Intriguingly, the ratio of dividing bPs to apical progenitors (radial glial cells) was found to be higher in the telencephalon than in the hindbrain (*Haubensak et al., 2004*). Similarly, recent studies revealed a higher ratio of dividing cells in the outer layers than in the lumen, when comparing the developing telencephalon to the lumbo-sacral spinal cord (*Yaghmaeian Salmani et al., 2018*). Albeit still limited in their scope, these studies suggest that a similar scenario is playing out along the A-P axis of mouse CNS as that observed in *Drosophila*, with an anteriorly extended phase of progenitor proliferation and a higher prevalence of proliferating daughter cells.

In addition to the similarities between *Drosophila* and mouse regarding progenitor generation, as well as progenitor and daughter cell proliferation, the genetic mechanisms controlling these events may also be conserved. Mouse orthologues of the *Drosophila* brain TFs studied herein that is Nr2E1/Tlx (Tll); Otp, Rax and Arx (Otp); Tbx2/3/6 (Doc1/2/3); and FezF1/2 (Erm), are restricted to the brain and are known to be critical for normal mouse brain development, and in several cases for promoting proliferation (*Eckler and Chen, 2014*; *Islam and Zhang, 2015*; *Kitamura et al., 2002*; *Lu et al., 2013*; *Price et al., 2009*; *Trowe et al., 2013*; *Wang and Lufkin, 2000*). Furthermore, Hox genes are not expressed in the mouse forebrain and there is a generally conserved feature of brain TFs expressed anteriorly and Hox genes posteriorly (*Arendt and Nübler-Jung, 1999*; *Holland, 2003*; *Philippidou and Dasen, 2013*; *Reichert and Simeone, 1999*; *Thor, 1995*). Mutation and misexpression has revealed that Hox genes are anti-proliferative also in the vertebrate CNS (*Economides et al., 2003*; *Yaghmaeian Salmani et al., 2018*). Moreover, PRC2 (*EED*) mouse mutants show extensive expression of Hox genes into the forebrain and reduced gene expression of for example Nr2E1, Fezf2 and Arx (*Yaghmaeian Salmani et al., 2018*). This is accompanied by reduced proliferation in the telencephalon and a microcephalic brain, while the spinal cord does not appear effected (*Yaghmaeian Salmani et al., 2018*).

Gene expression and phylogenetic consideration recently led to the proposal that the CNS may have evolved by 'fusion' of two separate nervous systems, the apical and basal nervous systems, present in the common ancestor (*Arendt et al., 2016*; *Nielsen, 2012*; *Nielsen, 2015*; *Tosches and Arendt, 2013*). Interestingly, in arthropods for example *Drosophila*, the brain and nerve cord initially form in separate regions only to merge during subsequent development (*Hartenstein and Campos-Ortega, 1984*). Recent studies of the role of the PRC2 complex and Hox genes in controlling A-P differences in CNS proliferation, in both *Drosophila* and mouse, lend support for the notion of a 'fused' CNS (*Yaghmaeian Salmani et al., 2018*). This idea is further supported by recent studies of the epigenomic signature and early embryonic cell origins of the anterior versus posterior developing CNS (*Metzis et al., 2018*). The findings outlined herein, showing that brain hyperproliferation is driven not only by the lack of Hox homeotic gene expression, but also by the specific expression of highly conserved brain TFs, lend further support to the notion of a separate evolutionary origin of brain and nerve cord.

It is tempting to speculate that the possibly separate evolutionary origins of the brain and nerve cord may manifest not only as distinct modes of neurogenesis, but also be reflected by separate regulatory mechanisms. These would involve brain TFs acting anteriorly, generating an abundance of progenitors, as well as driving progenitor and daughter cell proliferation. Conversely, Hox genes would act posteriorly, counteracting progenitor generation, as well as tempering progenitor and daughter cell proliferation. In this model, PRC2 would act as a 'gate keeper', ensuring that Hox genes are restricted from the brain and thereby promoting brain TF expression. This model clearly

represents an over-simplification, but may serve as a useful launching point for future comparative studies in many model systems.

## Materials and methods

### Fly stocks

#### Mutant stocks

*Df(3L)ED225* (Bloomington Drosophila Stock Center; BL#8081), a deletion of *grim*, *rpr*, *skl*, and most of the upstream region of *hid*. *esc$^5$* (BL#3142). *esc$^{21}$* (BL#3623). *esc$^{Df}$* = *Df(2L)Exel6030* (BL#7513). *esc* maternal/zygotic mutants were generated as previously described (*Yaghmaeian Salmani et al., 2018*). *erm$^{Df}$* = *Df(2L)Exel6006* (BL#8000)/*Df(2L)Exel8005* (BL#7779). *otp$^{Df}$* = *Df(2R)Exel7166* (BL#7998), which also removes *hbn* and *Rx*, as well as the genes: *Act57B*, *CG10543*, *CG15649*, *CG15650*, *CG15651*, *CG3216*, *CG33704*, *CG34115*, *CG9313*, *CG9344*, *Cib2*, *dgt3*, *lpk1*, and *Prosalpha3*. *tll$^{l49}$* (BL#7093). *tll$^{Df}$* = *Df(3R)Exel6217* (BL#7695). *Doc1,2,3$^{Df}$* triple mutants = *Df(3L)DocA*, which also removes *CG5087*, *CG5194*, *CG5144*, *Argk* and *CG4911* (*Reim et al., 2003*) (provided by Manfred Frasch, Friedrich-Alexander University, Erlangen, Germany) placed over *Df(3L)BSC130* (BL#9295).

#### Crispr/Cas9 mutants

Mutations were generated as previously described (*Stratmann and Thor, 2017*). Briefly, gRNAs flanking *otp*, *Rx* and *hbn* were designed with sequences:

> CTTCCACTTCGCGCCATCCC and CGCGGGAATAGACATCGGGG (Rx)
> CAAGCCAATTGATGCATGAC and ACGGCCTTTTTAGACGCACT (otp)
> CAAGCCAATTGATGCATGAC and ACGGCCTTTTTAGACGCACT (hbn)

gRNA constructs expressing tandem gRNAs, in vector pCFD4-U61-U63 (*Port et al., 2014*) (Addgene # 49411; provided by Simon Bullock, MRC, Cambridge, UK), were integrated into strains BL#9736 (genomic position 53B, for *Rx*), BL#9744 (genomic position 89E, for *otp*) and BL#9750 (genomic position 65B, for *hbn*) (BestGeneInc, Chino Hills, CA, USA), mediated by PhiC31 integrase driven by *vas-int* (*Bischof et al., 2007*). Gene deletion was triggered by crossing males of the transgenic tandem gRNA flies to virgins of *vas-Cas9* (BL#51323). Alleles identified were *Rx$^{ST1}$* and *otp$^{ST1}$*, initially identified by lethality when placed over the genomic deletion of the region (*Df(2R)Exel7166*). Sequencing of *Rx$^{ST1}$*, by PCR-cloning-Sanger-sequencing, revealed the predicted deletion. Sequencing of *otp$^{ST1}$*, by whole-genome-sequencing, revealed mutations at both gRNA target sites (see *Figure 1—source data 1* for details). For *hbn* no alleles were identified.

#### *Gal4* lines

*elav-Gal4* (*DiAntonio et al., 2001*) (provided by Aaron DiAntonio, Washington University, St.Louis, MO, USA). *pros-Gal4* on chromosome III (provided by Fumio Matsuzaki, RIKEN CBD, Kobe, Japan). *da-Gal4* on chromosome II and III (BL#55849). *vg-Gal4* = *vgBE-Gal4, UAS-flp; tubP > stop > Gal4, UAS-GFP* (*Crickmore and Mann, 2006*) (provided by Richard S. Mann, Columbia University, New York, USA). *wg-Gal4* (provided by Konrad Basler, University of Zurich, Switzerland). *OK107-Gal4* (BL#854).

#### *UAS* lines

*UAS-Ubx* (*Merabet et al., 2011*) (provided by Samir Merabet, IGFL, Lyon, France). *UAS-abd-A* and *UAS-Abd-B* (*Monedero Cobeta et al., 2017*). *UAS-Doc1*, *UAS-Doc2*, *UAS-Doc3* (*Reim et al., 2003*) (provided by Manfred Frasch, Friedrich-Alexander University, Erlangen, Germany). *UAS-tll* (Kyoto Drosophila Stock Center #109680). *UAS-ey* (BL#56560). *UAS-toy* (BL#6257). *UAS-nls-myc-EGFP* (referred to as *UAS-EGFP*) (*Allan et al., 2003*).

From FlyORF (*Bischof et al., 2013*): *UAS-hbn* (#F000105), *UAS-Rx* (#F000645), *UAS-optix* (#F000244), *UAS-otp* (#F000016), *UAS-ems* (#F000141). *UAS-erm* (22A) and *UAS-bsh* (53B) were generated by injecting the pGW-HA.attB DNA FlyORF constructs (provided by Johannes Bischof and Konrad Basler, University of Zurich, Switzerland) into landing site strains BL#9752 (22A) and

BL#9736 (53B) (BestGeneInc, Chino Hills, CA, USA). Transgene integration was mediated by X-chromosome expression PhiC31integrase driven by *vas-int* (*Bischof et al., 2007*).

## Marker lines

*9D11-lacZ* (*erm-lacZ*) (*Haenfler et al., 2012*) (provided by Cheng-Yu Lee, University of Michigan, Ann Arbor, MI, USA). *tll-EGFP* (BL#30874). *lbe(K)-EGFP* (*Ulvklo et al., 2012*). *wach = wor-Gal4 ase-Gal80; 20xUAS-6xCherry:HA* (*Álvarez and Díaz-Benjumea, 2018*) (provided by Fernando Diaz-Benjumea, Universidad Autonoma de Madrid, Madrid, Spain).

Mutants were maintained over *GFP*- or *YFP*-marked balancer chromosomes. As control *Oregon-R* or *Gal4/+* were used. Staging of embryos was performed according to Campos-Ortega and Hartenstein (*Campos-Ortega and Hartenstein, 1985*).

## Immunohistochemistry

Rabbit anti-Doc2 (*Reim et al., 2003*) (1:2,000) (provided by Manfred Frasch, Friedrich-Alexander University, Erlangen, Germany). Rabbit anti-Tll (1:500) (provided by Ralf Pflanz, MPI, Goettingen, Germany). Rabbit anti-PntP1 (1:1,000) (provided by James B. Skeath, Washington University, St Louis, MO, USA). Rabbit anti-Hbn (1:400), rabbit anti-Bsh (1:400) and guinea pig anti-Rx (1:1,000) (provided by Claude Desplan, NYU, New York, USA). Rabbit anti-Cas (provided by Ward Odenwald, NINDS, Bethesda, USA). Rabbit anti-Erm (1:50) and rabbit anti-Insc (1:1,000; provided by Wang Hongyan, Temasek Life Sciences, Singapore). Rabbit anti-phospho-histone H3-Ser10 (PH3) (1:1,000; #06–570; Upstate/Millipore, Billerica, MA, US). Rabbit anti-Abd-A (1:100) (provided by Maria Capovila, CNRS, Sophia Antipolis, France). Rabbit anti-CycE (1:500; #sc-33748; Santa Cruz Biotechnology, Santa Cruz, CA, USA). Rat anti-E2f1 (1:100), guinea pig anti-Dpn (1:1,000), and guinea pig anti-Dap (1:1,000) (*Baumgardt et al., 2014*). Rat anti-Dpn (1:500) (*Ulvklo et al., 2012*). Rat anti-Stg (1:500) (*Bivik et al., 2016*). Rat mAb anti-GsbN (1:10) (*Buenzow and Holmgren, 1995*) (Robert Holmgren, Northwestern University, USA). Mouse mAb anti-Mira (Fumio Matsuzaki, RIKEN CBD, Kobe, Japan). mAb anti-Ubx FP3.38 (1:10) (provided by Rob White, University of Cambridge, Cambridge, UK). Rabbit mAb α-PH3-Ser10 (1:1,000; Cat.no. ab177218), Chicken anti-GFP (1:2,000; cat.no. ab13970), rat mAb anti-PH3-Ser28 (1:1,000; Cat.no. ab10543; Abcam, Cambridge, UK). Rat mAb anti-Elav 7E8A10 (1:10), mouse mAb anti-Abd-B 1A2E9 (1:10), mouse mAb anti-Pros MR1A (1:10), mouse mAb anti-Repo 8D12 (1:10), mouse mAb anti-ß-gal JIE7 (1:10), mouse mAb anti-BP102 (1:10) (Developmental Studies Hybridoma Bank, Iowa City, IA, USA). Secondary antibodies used were FITC-, rhodamine RedX-, AMCA-, and Cy5/Alexa Fluor647-conjugated donkey antibodies (1:200; Jackson Immunoresearch Laboratories, West Grove, PA).

## Confocal imaging and data acquisition

Zeiss LSM700 or Zeiss LSM800 confocal microscopes were used in the acquisition of fluorescent images. Confocal stacks were merged or visualised using Fiji software (*Schindelin et al., 2012*). Images and graphs were compiled in Adobe Illustrator.

## Quantification of proliferation in brain gene mutants

Embryos were collected at St13, dissected and fixed in 4% PFA. Semi-automated macro, ImageJ macro language-based (*Yaghmaeian Salmani et al., 2018*) was used for the quantification of proliferating NBs and GMCs and NB total number in B1-B2 with the Fiji software.

## Quantification of total cell numbers

Embryos were dissected in PBS and fixed for 20 min in 4% PFA. After fixation, immunostaining was performed as previously described (*Baumgardt et al., 2014*). Embryos were mounted in Vectashield media (Vector Laboratories). Semi-automated macro, ImageJ macro language-based, was designed and used for the quantification of the volume of each segment with the Fiji software. The same software was used to manually count the number of cells and volumes of areas of nerve cord in order to estimate average single cell volume (*Yaghmaeian Salmani et al., 2018*) (see *Figure 2—source data 1* for details).

## Fluorescence intensity measurements

Experimental and control embryos of the same stage were dissected on the same slide to ensure identical staining conditions. The fluorescence intensity (mean pixel intensity x area occupied by the signal) of individual cells was measured using Fiji software on single confocal layers 1 µm thick, in the medial plane of the cells (*Yaghmaeian Salmani et al., 2018*).

## Acknowledgements

We are grateful to Simon Bullock, Wang Hongyuan, Fernando Diaz-Benjumea, Manfred Frasch, Cheng-Yu Lee, Aaron DiAntonio, Rob White, Gert Technau, Fumio Matsuzaki, Richard S Mann, Claude Desplan, James B Skeath, Ward Odenwald, Johannes Bischof, Konrad Basler, Samir Merabet, Maria Capovila, Robert Holmgren, Ralf Pflanz, the Developmental Studies Hybridoma Bank at the University of Iowa, FlyORF, and the Kyoto and Bloomington Stock Centers for sharing reagents and advice. We thank Simon Sprecher for critically reading the manuscript. Johannes Stratmann, Helen Ekman, Carolin Jonsson and Annika Starkenberg provided excellent technical assistance.

## Additional information

### Funding

| Funder | Grant reference number | Author |
|---|---|---|
| Knut och Alice Wallenbergs Stiftelse | KAW2011.0165 | Stefan Thor |
| Vetenskapsrådet | 621-2013-5258 | Stefan Thor |
| Cancerfonden | 140780 | Stefan Thor |
| Knut och Alice Wallenbergs Stiftelse | KAW2012.0101 | Stefan Thor |
| Cancerfonden | 150663 | Stefan Thor |

The funders had no role in study design, data collection and interpretation, or the decision to submit the work for publication.

### Author contributions

Jesús Rodriguez Curt, Conceptualization, Data curation, Formal analysis, Investigation, Visualization, Methodology, Writing—original draft, Writing—review and editing; Behzad Yaghmaeian Salmani, Conceptualization, Data curation, Formal analysis, Investigation, Visualization, Writing—original draft, Writing—review and editing; Stefan Thor, Conceptualization, Resources, Supervision, Funding acquisition, Visualization, Writing—original draft, Writing—review and editing

### Author ORCIDs

Jesús Rodriguez Curt https://orcid.org/0000-0002-5712-3031
Behzad Yaghmaeian Salmani https://orcid.org/0000-0002-4221-6243
Stefan Thor https://orcid.org/0000-0001-5095-541X

### Decision letter and Author response

Decision letter https://doi.org/10.7554/eLife.45274.026
Author response https://doi.org/10.7554/eLife.45274.027

## Additional files

### Supplementary files

• Source data 1.
DOI: https://doi.org/10.7554/eLife.45274.023
• Transparent reporting form

DOI: https://doi.org/10.7554/eLife.45274.024

**Data availability**

All data generated or analysed during this study are included in the manuscript and supporting files.

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
