## [Decision Letter]

Thank you for submitting your article "Anterior CNS expansion driven by brain transcription factors" for consideration by *eLife*. Your article has been reviewed by three peer reviewers, and the evaluation has been overseen by K VijayRaghavan as the Senior and Reviewing Editor. The following individuals involved in review of your submission have also agreed to reveal their identity: Hongyan Wang (Reviewer #2) and Rita Sousa-Nunes (Reviewer #3).

The reviewers have discussed the reviews with one another and the Reviewing Editor has drafted this decision to help you prepare a revised submission.

Summary:

In this and older work, the authors investigate a feature of the nervous system that is so common that it seems to have evaded systematic investigation: that most extant brain are anteriorly enlarged. They earlier showed that the anterior brain expansion was due to increased neuroblast (NB) and daughter cell proliferation and that polycomb repressor complex expression in the anterior region ensured that hox genes, which reduce proliferation in the VNC, were excluded from the brain, allowing it to expand.

Here, the authors explore the mechanistic details underlying this phenomenon. Their approach is an overexpression study. Shortlisting 14 transcription factors expressed in the brain, they identified four – Tll, Otp/Rx/Hbn, Doc1/2/3, and Erm – whose misexpression in the CNS resulted extended proliferation of NBs and daughter cells. They show that these 'brain TFs' can repress Hox genes and can rescue the PRC mutant effect of reduced cell proliferation in the brain. Strikingly, misexpressing these TFs (2 of them) in the wing disc results in neural specific markers – dpn, pros, ase, BP102, cas, repo and elav – being upregulated in the wing disc.

Essential revisions:

1) While substantial work has gone into this study, one important concern is that it relies on over-expression of potent transcription factors in combinations that may not occur in the normal developing brain. The misexpression approach is a great way to identify candidates. However, could the authors root the rest of the analyses on native expression patterns? What proportion of the B1 NBs express each of these TFs, combinations of these TFs, and all four of these TFs? Without this, it's not clear why they decided to misexpress all 4 factors, or why they picked the specific 2 factors for further misexpression studies. If the claim is that the co-expression of these factors in the anterior brain has allowed it to expand, such an analysis would be essential.

2) As it stands, the effects of their co-misexpressions are hard to interpret. For example, the graph in Figure 2C seems to suggest that amongst the 2-factor combinations the erm/doc2 combination has the strongest effect on dividing daughters, and the tll/otp combination on dividing NB. (The 4 factor combination has the strongest effect on both these.) Yet the authors selected the tll/erm combination in the rest of the study, which has only a modest increase in either's proliferation. What this particular combination does show is an increase in TOTAL number of NB and a potent ability to specify NBs in ectopic tissue. This is especially evident in the wing disc, which is possibly the most striking result of this paper. Adding the other two factors to this combination seems to suppress this ability! The authors should discuss this.

3) The above brings up the question of specification vs. proliferation, which also comes up in their loss of function analysis where the authors note that total numbers of NBs are reduced. Ideally these aspects would need to be delineated in their analyses. (To start with, maybe the reduction in proliferating NBs could be represented as a percentage of total NBs in each case.) However, given these are misexpression studies of combinations of TFs, it is not certain whether this could actually be done meaningfully. For example, there must be a complex interaction between tll and erm: tll-/- have dramatically reduced NBs, erm-/- have a massive expansion of NBs and even short pulses of it is known to reduce NB numbers. (Both are expressed in type II NBs, though we are not aware if they are in the same cells). Yet misexpressing both results in the ectopic specification of NBs, (and misexpressing all four suppresses it). This too needs to be discussed.

4) For the above three points the authors should revisit their data, and quantify the percentage of B1 NBs that co express Tll and Erm, and the percentage of brain NBs that co-express all four.

5) In Figure 1—figure supplement 1, there seemed to be more PH3+ cells in E than in M or O. Please provide quantifications to support the conclusion that J-O have stronger proliferation effect than B-I.

6) Does misexpression of key PRC2 components also lead to VNC-to-brain or discs-to-brain reprogramming? If this experiment will take the authors more than two months to re-submit, please address this in the Discussion.

7) Does PRC2 complex directly regulate the expression of these four brain TFs identified in this study? If this experiment will take the authors more than two months to re-submit, please address this in the Discussion.

8) Provide a further explanation of the criteria used to select the 14 TFs focused on. In the Introduction the authors say they "applied a number of criteria to focus in on 14 transcription factors (TFs) specifically expressed in the developing embryonic brain". Then in the first section of the Results the authors explain that they surveyed publicly available information of TFs expressed in the *Drosophila* brain, reaching 14 TFs that they collectively term "brain TFs"; however, there are more TFs than these expressed in the brain, which should be made clear in the writing, as should be, therefore, why those specific 14 were selected for functional analyses.

9) Of the 14 "brain TFs", the authors then say they focused on "four genes" but in fact they are four "gene families" – please amend; and make it explicit in the loss-of-function experiments of otp/Rx/hbn and Doc1/2/3 gene families that the deficiencies employed result in deletion of additional genes (say which). Based on the results depicted in Figure 1—figure supplement 1 (ability to induce prolonged proliferation in late embryonic VNC) the authors selected otp, erm, tll and Doc1-3 for further study; please explain why bsh was rejected when the effect depicted (panel E) appears at least as strong as that of the Doc1-3 genes.

10) In the Results section "Combinatorial brain TF misexpression drives nerve cord proliferation" the authors state that "Because elav-Gal4 is a late driver, activated in NBs subsequent to their delamination (Berger et al., 2007), the supernumerary NBs are unlikely to stem from extra NBs generated during delamination. Rather, this effect may reflect aberrant symmetric NB divisions." This is an important mechanistic point that could be but was not addressed directly. Please provide quantification of NBs at St13 to confirm that no more NBs are present then (i.e., no extra have delaminated); and investigate whether NB symmetric divisions and/or evidence of daughter cell reversion to NBs can be found.

11) The description of effects of mutations on cell cycle factors is poor. Please spell out findings clearly and include in Figure 2—figure supplement 4 a schematic of where in the cell cycle each of the genes analyzed acts, as well as a schematic for each of the mutants showing how each of the cell-cycle phases is altered (presumably, if overall cell-cycle length is unchanged, lengthening of (a) cell-cycle stage(s) is compensated by shortening of (a)nother(s)). Furthermore, does the quantitative data in this figure pertain solely to the NB cell cycle? This was not said and is an important point.

12) In the Results section "Combinatorial brain TF misexpression reprograms nerve cord to brain" the authors say: "We conclude that while UAS-tll,erm co-misexpression is sufficient to trigger ectopic CNS development in the ectoderm". Could the statement be a bit more specific, i.e., are the results compatible with reprogramming into brain, specifically into type II lineages? (one of which, DL1, generates glia…)? This idea merits direct testing as it would provide additional insight into the authors' observations. In wing discs, co-expression of Dpn and Ase was surveyed (Figure 5—figure supplement 4) precisely to ascertain whether NBs were of type I or II – what about Ase in the ectopic VNC NBs? And can evidence for ectopic INP-like cells be found? Please quantify Dpn+/Ase+, Dpn+/Ase- and Dpn-/Ase+ in order to make a quantitative statement about this in both the UAS-tll,erm vs. UAS-tetra genotypes in the VNC and wing discs.

13) Figure 4H and Figure 5—figure supplements 3 and 4: It is strange that the authors considered Pros not to be induced by UAS-tetra when Elav was, given that abundance of the two markers seems analogous (if the pictures shown are representative) and that Pros is necessary for neuronal specification in *Drosophila*. (As few Repo+ cells are shown to be induced by UAS-tll,erm, which the authors considered able to induce Repo). Presumably Pros is expressed but at very low levels – the authors could confirm this (e.g., by FISH or WB). Related to this, it is puzzling that Figure 5—figure supplement 4 legend says that "(E-F) vg-Gal4>UAS-tll,erm or triggers ectopic expression of Dpn, Elav and Repo, while UAS-Tetra only activates Dpn." whereas in Figure 4H UAS-Tetra is indicated as able to induce Elav (and a few Elav+ cells are shown in panel F).

14) In Figure 5 the authors are talking about the action of TFs in NBs, we believe, so it would make sense to look at cross regulation between these and Hox genes specifically in NBs and not throughout the field of neural cells as was done. Please quantify specifically in NBs (as was done for reciprocal experiment shown in Figure 6—figure supplement 1 – the content of which could be moved to main Figure 5). In Figure 6—figure supplement 1 – we just did not understand why Bsh was quantified in GMCs and not NBs – please explain.

15) Please show data points overlaid onto all histograms and specify in figure legends t-test significance (p value) corresponding to each of the number of asterisks indicated. In some cases (e.g. Figure 1I, Figure 5J) it is surprising that the results come up as statistically significantly different from controls and reviewers should have been given the raw data with the first submission rather than be told that authors "will upload all numerical data upon a possible re-submission".

---

## [Author Response]

Essential revisions:1) While substantial work has gone into this study, one important concern is that it relies on over-expression of potent transcription factors in combinations that may not occur in the normal developing brain. The misexpression approach is a great way to identify candidates. However, could the authors root the rest of the analyses on native expression patterns? What proportion of the B1 NBs express each of these TFs, combinations of these TFs, and all four of these TFs? Without this, it's not clear why they decided to misexpress all 4 factors, or why they picked the specific 2 factors for further misexpression studies. If the claim is that the co-expression of these factors in the anterior brain has allowed it to expand, such an analysis would be essential.

Because we do not have access to antibody and/or transgenic markers for all TFs under study herein, the justification for combining the 4 and 2 TFs is based upon a combination of expression analysis and phenotypes. Regarding expression analysis, we have now quantified the number of NBs expressing each TF, and also the number of NBs showing co-expression, and have added these data to Figure 2—figure supplement 1 (new graph, Figure 2—figure supplement 1O). For the 4 TF combination, we observe expression in Type II lineages of tll and Rx (NBs), and erm (INPs), and phenotypes for all 4 TFs/families; tll, otp/Rx/hbn, erm and Doc1/2/3. Hence, we would argue that all TFs studied herein are expressed in Type II NB lineages. We also observe tll in most, if not all Type I NBs, and Hbn, Doc2 and erm in a number of Type I NBs. In addition, we observe strong global NB and daughter proliferation effects (Figure 1G-H). Hence, we believe that the 4 TFs/families also are co-expressed in a large number of Type I NBs.

2) As it stands, the effects of their co-misexpressions are hard to interpret. For example, the graph in Figure 2C seems to suggest that amongst the 2-factor combinations the erm/doc2 combination has the strongest effect on dividing daughters, and the tll/otp combination on dividing NB. (The 4 factor combination has the strongest effect on both these.) Yet the authors selected the tll/erm combination in the rest of the study, which has only a modest increase in either's proliferation. What this particular combination does show is an increase in TOTAL number of NB and a potent ability to specify NBs in ectopic tissue. This is especially evident in the wing disc, which is possibly the most striking result of this paper. Adding the other two factors to this combination seems to suppress this ability! The authors should discuss this.

In the VNC, adding all 4 TFs together (UAS-Tetra) appears to be the strongest combination, which we think is a logical combinatorial result (Figure 2C). Regarding the wing disc, we have tested the potency of a number of additional singles, doubles, and triples in the wing disc, and added these to the results (new, Figure 5—figure supplements 1 and 2). We find that the tll, erm 2-TF combination is indeed the strongest in the wing disc. We are not sure why all other single, double, triple and tetra combinations are weaker than this particular double combination in the heterologous setting. One possible explanation is that the embryonic neuroectoderm is already primed for the generation of NBs. We have added a section to the Discussion on this (subsection “Brain TFs promote anterior CNS expansion by driving super-generation of NBs”, sixth paragraph).

3) The above brings up the question of specification vs. proliferation, which also comes up in their loss of function analysis where the authors note that total numbers of NBs are reduced. Ideally these aspects would need to be delineated in their analyses. (To start with, maybe the reduction in proliferating NBs could be represented as a percentage of total NBs in each case.) However, given these are misexpression studies of combinations of TFs, it is not certain whether this could actually be done meaningfully. For example, there must be a complex interaction between tll and erm: tll-/- have dramatically reduced NBs, erm-/- have a massive expansion of NBs and even short pulses of it is known to reduce NB numbers. (Both are expressed in type II NBs, though we are not aware if they are in the same cells). Yet misexpressing both results in the ectopic specification of NBs, (and misexpressing all four suppresses it). This too needs to be discussed.

Previous studies of erm function in the larvae found that erm mutants displayed more Type II NBs. Larval MARCM clone induction and marker analysis demonstrate that this is due to de-differentiation of INPs back to type II NBs, rather than excess generation of Type II NBs in the embryo (Weng et al., 2010). We do not find extra Type II or Type I NBs in erm mutants (Figure 1O). To address if there is de-differentiation of INPs back to NBs we have now quantified the number of NBs (Dpn numbers) inside wach (Type II) lineages in erm mutants (new graph, Figure 1Q). We do not find evidence for extra NBs inside the Type II lineages in erm mutants. However, erm mutants do show reduced number of cells generated in the embryonic Type II lineages, showing that it is important for lineage progression. Hence, the role of erm appears to be different in the embryonic versus larval Type II lineages. We have added a section on this to the Discussion (subsection “Brain TFs promote anterior CNS expansion by driving super-generation of NBs”, last paragraph).

tll mutants display reduction of both Type II and I NBs, and reduced proliferation. Hence, tll promotes NB generation, and tll and erm NB lineage progression. Against this backdrop of embryonic function, we do not find it surprising that misexpressing either one of them alone in the VNC triggers aberrant NB and daughter cell proliferation (Figure 2C).

Regarding the Tetra, it is the strongest UAS combo in the VNC, when driven from elav-Gal4 (Figure 2C-D), and also stronger in the embryonic ectoderm than the tll, erm double, when driven from da-Gal4. But in the wing disc the Tetra is weaker than the tll, erm double. We have no good explanation for the dichotomy between the VNC and the wing disc, but it may pertain to that the wing disc it not a neurogenic tissue while ventral ectoderm is already primed for NB generation, and it is possible that the competence of the two tissues make the UAS combinations act differently, rendering either double or tetra the most potent in wing disc or VNC respectively. We have added a section in the Discussion on this (see the sixth paragraph of the aforementioned subsection).

4) For the above three points the authors should revisit their data, and quantify the percentage of B1 NBs that co express Tll and Erm, and the percentage of brain NBs that co-express all four.

We have now quantified the number of NBs expressing each TF, and also the number of NBs showing co-expression, and have added these data to Figure 2—figure supplement 1 (new panel, Figure 2—figure supplement 1O).

5) In Figure 1—figure supplement 1, there seemed to be more PH3+ cells in E than in M or O. Please provide quantifications to support the conclusion that J-O have stronger proliferation effect than B-I.

We have now quantified the proliferation effects, and find that the 4 TFs chosen for further study indeed display the strongest effects in the VNC (new graph, Figure 1—figure supplement 1P).

6) Does misexpression of key PRC2 components also lead to VNC-to-brain or discs-to-brain reprogramming? If this experiment will take the authors more than two months to re-submit, please address this in the Discussion.

We have previously misexpressed, from UAS, a number of different PRC1 and PRC2 components, both wt versions and dominant-negatives. These include Sfmbt, Sfmbt-DMBT1, Scm, Sce, esc, Pho, Pho-N-term and Pho-C-term. We have however not observed any clear effects from these transgenes, driven by pros-Gal4 and/or elav-Gal4, as visualized by Hox gene expression, H3K27me3 intensity, VNC-marker expression and/or proliferation. There are published effects from misexpressing/overexpressing PcG components in the developing imaginal discs. However, in these disc experiments expression is driven for several days, while in the embryo we were scoring for GOF effects just hours after UAS onset. In the mouse, when we delete Eed (esc) it takes 2-3 embryonic days before we see a clear loss of H3K27me3. Hence, we think that PRC2 cannot easily be targeted by zygotic transgenic expression within the timeframe of *Drosophila* embryonic development.

7) Does PRC2 complex directly regulate the expression of these four brain TFs identified in this study? If this experiment will take the authors more than two months to re-submit, please address this in the Discussion.

We have previously found that Doc2 and Tll are downregulated in esc mutants (Yaghmaeian et al., 2018). We have now quantified Rx expression in esc mutants, and find significant downregulation also of Rx (new graph; Figure 6O). Regarding direct regulation, previous studies have addressed, by ChIP-seq, the genome-wide H3K27me3 profile in embryonic mesoderm. We find restricted H3K27me3 peaks over tll, Doc1/2/3, otp/Rx/hbn and erm. We can add images of these profiles to the supplementary data if the reviewers find this valuable.

8) Provide a further explanation of the criteria used to select the 14 TFs focused on. In the Introduction the authors say they "applied a number of criteria to focus in on 14 transcription factors (TFs) specifically expressed in the developing embryonic brain". Then in the first section of the Results the authors explain that they surveyed publicly available information of TFs expressed in the Drosophila brain, reaching 14 TFs that they collectively term "brain TFs"; however, there are more TFs than these expressed in the brain, which should be made clear in the writing, as should be, therefore, why those specific 14 were selected for functional analyses.

We have added further clarification to the selection process to the first Results section (subsection “Misexpression of brain TFs drives nerve cord proliferation”).

9) Of the 14 "brain TFs", the authors then say they focused on "four genes" but in fact they are four "gene families" – please amend; and make it explicit in the loss-of-function experiments of otp/Rx/hbn and Doc1/2/3 gene families that the deficiencies employed result in deletion of additional genes (say which). Based on the results depicted in Figure 1—figure supplement 1 (ability to induce prolonged proliferation in late embryonic VNC) the authors selected otp, erm, tll and Doc1-3 for further study; please explain why bsh was rejected when the effect depicted (panel E) appears at least as strong as that of the Doc1-3 genes.

We have changed to two genes (tll and erm) and two gene families (otp/Rx/hbn and Doc1/2/3) throughout. We have described that the deletions used that removes otp/Rx/hbn or Doc1/2/3 also removes other genes (Materials and methods, subsection “Mutant stocks”). We have now quantified the effects in the AFT VNC, when driven from elav-Gal4, and find that the four TFs chosen for further study indeed display the strongest effects in the VNC, and hence the image chosen for bsh was an outlier (new graph, Figure 1—figure supplement 1P).

10) In the Results section "Combinatorial brain TF misexpression drives nerve cord proliferation" the authors state that "Because elav-Gal4 is a late driver, activated in NBs subsequent to their delamination (Berger et al., 2007), the supernumerary NBs are unlikely to stem from extra NBs generated during delamination. Rather, this effect may reflect aberrant symmetric NB divisions." This is an important mechanistic point that could be but was not addressed directly. Please provide quantification of NBs at St13 to confirm that no more NBs are present then (i.e., no extra have delaminated); and investigate whether NB symmetric divisions and/or evidence of daughter cell reversion to NBs can be found.

Regarding “quantification of NBs at St13”, we already displayed these results in Figure 2H, showing more NBs. We have now looked for evidence of symmetric NBs divisions in the embryo VNC, focusing specifically on the row 5 NBs, using the wg-Gal4 driver. Intriguingly, we find evidence of symmetric NB divisions (new Figure 4—figure supplement 6). Symmetric NB divisions are reminiscent of the symmetric divisions of early neuroectodermal cells observed in brain regions.

11) The description of effects of mutations on cell cycle factors is poor. Please spell out findings clearly and include in Figure 2—figure supplement 4 a schematic of where in the cell cycle each of the genes analyzed acts, as well as a schematic for each of the mutants showing how each of the cell-cycle phases is altered (presumably, if overall cell-cycle length is unchanged, lengthening of (a) cell-cycle stage(s) is compensated by shortening of (a)nother(s)). Furthermore, does the quantitative data in this figure pertain solely to the NB cell cycle? This was not said and is an important point.

We have added a schematic depicting the role in the cell cycle of the cell cycle factors analysed (Figure 2—figure supplement 4L). We have clarified the effects (subsection “Combinatorial brain TF misexpression drives nerve cord proliferation”, last paragraph). We have also clarified in the text that we are measuring protein levels in NBs. Regarding adding a schematic for how the cell cycle phase is altered in each mutant and in the misexpression we struggled with this. For instance, erm positively regulates two “driver” cell cycle factors (CycE and E2f1) but negatively regulates another one (Stg). We are not sure what the outcome of this would be regarding where in the cell cycle erm mutants would be delayed/stopped. The same applies to several other LOF and GOF. It may be possible to address this by BrdU/EdU/PH3 labelling, but given the number of LOF and GOF backgrounds involved it would be outside of the scope of a resubmission.

12) In the Results section "Combinatorial brain TF misexpression reprograms nerve cord to brain" the authors say: "We conclude that while UAS-tll,erm co-misexpression is sufficient to trigger ectopic CNS development in the ectoderm". Could the statement be a bit more specific, i.e., are the results compatible with reprogramming into brain, specifically into type II lineages? (one of which, DL1, generates glia…)? This idea merits direct testing as it would provide additional insight into the authors' observations. In wing discs, co-expression of Dpn and Ase was surveyed (Figure 5—figure supplement 4) precisely to ascertain whether NBs were of type I or II – what about Ase in the ectopic VNC NBs? And can evidence for ectopic INP-like cells be found? Please quantify Dpn+/Ase+, Dpn+/Ase- and Dpn-/Ase+ in order to make a quantitative statement about this in both the UAS-tll,erm vs. UAS-tetra genotypes in the VNC and wing discs.

We have now stained the VNC for Ase/Dpn in da>double and da>Tetra, as well as quantified the previous vg>double and vg>Tetra wing disc data. Indeed, we find evidence for that both the erm,tll and Tetra co-misexpression can trigger ectopic generation of Type II-like NBs. We have added these new data to the manuscript (new Figure 4 and new graphs in Figure 5—figure supplement 4G-I). We are also discussing this in the Results (subsection “Combinatorial brain TF misexpression reprograms the nerve cord to a brain like CNS”, second paragraph and subsection “Combinatorial brain TF misexpression reprograms wing disc to a brain-like CNS”, third paragraph) and Discussion (subsection “Brain TFs promote anterior CNS expansion by driving super-generation of NBs”, fifth paragraph and subsection “Brain TFs promote anterior CNS expansion by driving NB and daughter cell proliferation”, second paragraph).

13) Figure 4H and Figures 5—figure supplements 3 and 4: It is strange that the authors considered Pros not to be induced by UAS-tetra when Elav was, given that abundance of the two markers seems analogous (if the pictures shown are representative) and that Pros is necessary for neuronal specification in Drosophila. (As few Repo+ cells are shown to be induced by UAS-tll,erm, which the authors considered able to induce Repo). Presumably Pros is expressed but at very low levels – the authors could confirm this (e.g., by FISH or WB). Related to this, it is puzzling that Figure 5—figure supplement 4 legend says that "(E-F) vg-Gal4>UAS-tll,erm or triggers ectopic expression of Dpn, Elav and Repo, while UAS-Tetra only activates Dpn." whereas in Figure 4H UAS-Tetra is indicated as able to induce Elav (and a few Elav+ cells are shown in panel F).

We apologize for these imprecise statements. UAS-Tetra works very poorly in the wing discs, but can indeed trigger limited ectopic expression of Dpn, Pros, Elav, Ase, Mira and Cas, but not Insc, Repo and BP102. We have changed the summary table (Figure 5H) and the Results and figure legend texts accordingly.

14) In Figure 5 the authors are talking about the action of TFs in NBs, we believe, so it would make sense to look at cross regulation between these and Hox genes specifically in NBs and not throughout the field of neural cells as was done. Please quantify specifically in NBs (as was done for reciprocal experiment shown in Figure 6—figure supplement 1 – the content of which could be moved to main Figure 5). In Figure 6—figure supplement 1 – we just did not understand why Bsh was quantified in GMCs and not NBs – please explain.

Ubx, Abd-A and Abd-B expression in brain TF misexpression was quantified in NBs (Figure 6J). Rx and Hbn were quantified in NBs, while Bsh was quantified in all cells. Now we have also quantified Bsh in NBs (Figure 6—figure supplement 1G).

15) Please show data points overlaid onto all histograms and specify in figure legends t-test significance (p value) corresponding to each of the number of asterisks indicated. In some cases (e.g. Figure 1I, Figure 5J) it is surprising that the results come up as statistically significantly different from controls and reviewers should have been given the raw data with the first submission rather than be told that authors "will upload all numerical data upon a possible re-submission".

We have replaced all bar graphs with scatter plots. We have specified what the asterisks correspond to. We also provide an excel file with all raw numbers underlying the graphs. The reason for significances, or lack thereof, may be easier to interpret from the new scatter plots.